# A murine model lacking *Lyst* recapitulates Chediak-Higashi syndrome with an earlier-onset neurodegenerative phenotype

Sunny Greene [1], Mackenzie L. Talbert[1], F. Graeme Frost[2], Patricia M. Zerfas[3], Danielle Springer[4], Audrey Noguchi[4], Marie Morimoto[2], Dawn Maynard[1], Lisa Garrett[5], Gene Elliot[5], Maria Traver [6], David Yarnell[1], Petcharat Leoyklang[2], John D. Burke[1], Elena-Raluca Nicoli [2], William A. Gahl[1,2], Wendy J. Introne [1] & May Christine V. Malicdan [1,2] ✉

Chediak-Higashi syndrome (CHS) is a rare, autosomal recessive disorder caused by pathogenic variants in the lysosomal trafficking regulator (*LYST*) gene and characterized by significant immunological and neurological impairment. Current mouse models do not replicate the early-onset neurological symptoms of patients. We develop and characterize a CHS model lacking the *Lyst* gene (ΔLYST-B6) using CRISPR-Cas9. The ΔLYST-B6 mouse exhibits key CHS features, including partial oculocutaneous albinism, prolonged bleeding, and enlarged intracellular granules. Molecular analyses confirm LYST deficiency, with reduced Lyst mRNA and protein levels across various tissues. Histological examination reveals progressive cerebellar Purkinje cell loss and axonal degeneration in peripheral nerves. Importantly, the ΔLYST-B6 show significant neurological impairment by 6 months of age. Lipidomic and transcriptomic analyses highlight increased proinflammatory lipid levels and immune signaling, suggesting neuroinflammation in CHS pathology. The ΔLYST-B6 mouse provides a valuable tool for studying the underlying mechanisms of CHS-associated neurodegeneration and for developing potential therapeutic strategies.

Chediak-Higashi syndrome (CHS, OMIM: 214500) is a rare, autosomal recessive disease with congenital immunodeficiency, bleeding diathesis, partial oculocutaneous albinism, and progressive neurologic impairment that include intellectual disability, parkinsonism, ataxia, and sensorimotor neuropathies[1–3]. The age of onset and the severity of the clinical presentation vary considerably. While allogeneic bone marrow or hematopoietic stem cell transplantation is a curative treatment for the immunodeficiency, the central and peripheral neurological deterioration is a consistent manifestation for all patients with no effective therapies[4]. Bi-allelic pathogenic loss-of-function variants in the lysosomal trafficking regulator (*LYST*) gene, which encodes a 429-kDa protein, have been identified as the cause of

CHS[5–7]. Despite the recognition of *LYST* variants as the cause of CHS nearly three decades ago, the precise function of LYST and how its impairment cause CHS remain unknown. This knowledge gap is likely due to several factors, including the absence of an animal model that recapitulates all aspects of CHS.

Current mouse models used to study CHS recapitulate several aspects of CHS but notably lack a robust neurological phenotype (Table 1). The most utilized model, the *Lyst^{bg-J}* or the *beige* mouse, has a well characterized immunodeficiency[8,9], but only displays a subtle behavioral phenotype at 12 months old and Purkinje cell loss late in life[10]. The *Lyst^{bg-J}* allele backcrossed onto the DBA/2J background (D2.*Lyst*) accelerated the severity of

[1]Human Biochemical Genetics Section, Medical Genetics Branch, National Human Genome Research Institute, National Institutes of Health, Bethesda, MD, USA. [2]National Institutes of Health Undiagnosed Diseases Program, National Human Genome Research Institute, National Institutes of Health, Bethesda, MD, USA. [3]Diagnostic and Research Services Branch, Office of Research Services, National Institutes of Health, Bethesda, MD, USA. [4]Murine Phenotyping Core, National Heart Lung & Blood Institute, National Institutes of Health, Bethesda, MD, USA. [5]Embryonic Stem Cell and Transgenic Mouse Core, National Human Genome Research Institute, National Institutes of Health, Bethesda, MD, USA. [6]Twinbrook Imaging Facility, Laboratory of Immunogenetics, National Institute of Allergy and Infectious Diseases, National Institutes of Health, Bethesda, MD, USA. ✉e-mail: maychristine.malicdan@nih.gov

**Table 1 | Phenotypic comparison of the ΔLYST allele on the C57BL/6J and DBA/2J backgrounds, *beige* allele on the C57BL/6J and DBA/2J backgrounds, and Lyst[Ing3618] allele on the C57BL/6J background**

| Phenotype | *Lyst[bg-J]* C57BL/6 J | *Lyst[bg-J]* DBA/2 J | *Lyst[Ing3618]* C57BL/6 J | ΔLYST-B6 C57BL/6 J | ΔLYST-DBA DBA/2 J |
|---|---|---|---|---|---|
| Partial albinism | ✓ | ✓ | ✓ | ✓ | ✓ |
| Ophthalmic phenotype (retina or iris)[a] | ✓ | ✓ | × | ✓ | Not tested |
| Hematological phenotype | ✓ | Not tested | ✓ | ✓ | Not tested |
| Immunological phenotype | ✓ | Not tested | × | Not tested | Not tested |
| Onset of neurological phenotype | >12 months (by CPSS) | >12 months (by CPSS) | 12 months (by coat hanger and stationary beam test) | 6 months (by composite ataxia score) | 3 months (by composite ataxia score) |
| Onset of significant Purkinje cell loss | 17-20 months | 17–20 months | 12 months[1] | 18 months | Not tested |
| Onset of significant peripheral neuropathy | Not tested | Not tested | 12 months | 3 months | Not tested |
| Reference | Hedberg-Buenz et al.[10] | Hedberg-Buenz et al.[10], Tantrow et al. 2010 | Rudelius et al.[12] | This study | This study |

✓ present, × absent, *CPSS* composite phenotypic scoring system for mouse models of cerebellar ataxia (Guyenet et al. 2010); [1]Intracytoplasmic inclusions were observed by histological analysis starting at 5 months in the homozygous mutant mice. [a]Note, most studies focused on characterizing the retina, except in Tantrow et al, where the study focused on the iris.

the neurological phenotype to some degree but only at 12 months of age[10,11]. Another model, the *Lyst[Ing3618]* (Granny mouse) with a missense mutation in a highly conserved region of the *Lyst* WD40 protein domain[12], presents with only a moderate neurological phenotype including Purkinje cell loss and peripheral neuropathy at 12 months.

To fill this gap, we generated a knockout mouse model (ΔLYST-B6) using CRISPR-Cas9 gene editing. Molecular, phenotypic, behavioral, and histological characterization identified significant neurological impairment in the ΔLYST-B6 model that better reflects the CHS-associated neurodegeneration. Lipidomic and transcriptomic analyses of the brain identified increased abundance of proinflammatory lipids and enrichment of immune signaling pathways.

This comprehensive characterization of this mouse model indicates that the ΔLYST-B6 recapitulates the features of the disease with a robust neurological phenotype and guides our understanding of the pathophysiology of CHS. Moreover, this ΔLYST-B6 model could be suitable for expanding approved therapeutic approaches for CHS patients to include treatments targeting neurological manifestations of the disorder in addition to the complex immunodeficiency and albinism phenotype.

## Results

### Generation and molecular validation of LYST-deficient mice

The murine *Lyst* gene contains 53 exons and shares a high homology to the human *LYST* gene, with 85% protein similarity[13]. We utilized two single CRISPR/Cas9 guide RNA sites targeting exons 4 and 53 of *Lyst* to delete a 149 kb segment (Fig. 1a, chr13:13,630,083_13,778,965del) (official name: C57BL/6J-*Lyst*[em1Mldn/Mldn])Animal genotyping by multiplex PCR demonstrated a 447 bp band in WT mice, a 642 bp band in ΔLYST-B6 mice, and both bands in heterozygous mice (Fig. 1b, Supplementary Table 1b).

Quantitative PCR revealed the absence of *Lyst* mRNA using a TaqMan assay amplifying exons 52-53 boundary in all ΔLYST-B6 tissues tested (Fig. 1c). *Lyst* mRNA expression at the exons 1-2 boundary, upstream of the CRISPR-Cas9 deleted region, showed comparable expression between WT and ΔLYST-B6 in forebrain and cerebellum (Fig. 1c, $P \geq 0.046$, $\alpha = 0.005$), but were reduced in liver, spleen, and kidney (Fig. 1c, $P \leq 8.42 \times 10^{-5}$). LYST protein was undetectable in all ΔLYST-B6 tissues tested and MEFs (Fig. 1d, Supplementary Fig. 1). These results confirm that the ΔLYST-B6 is a global loss-of-function model with reduced *Lyst* mRNA and LYST protein across various tissues.

### Phenotypic characterization of the ΔLYST-B6 mouse model

ΔLYST-B6 mice phenotypically present with a grey coat and lighter skin compared to the black coat and darker skin of WT mice, becoming evident two weeks after birth (Fig. 2a). Histological analysis of the whole eyeball revealed altered pigmentation in the retinal pigment epithelium (RPE) and choroid of the ΔLYST-B6 mouse retina (Fig. 2b).

ΔLYST-B6 mice exhibited prolonged bleeding time following tail tip transection. While bleeding stopped at an average of 34.4 seconds in the WT mice, ΔLYST-B6 mice bled continuously for 20 min ($P = 7.60 \times 10^{-9}$, Fig. 2c). Whole-mount platelet electron microscopy revealed a reduction of dense granules of ΔLYST-B6 mice (Fig. 2d). Peripheral blood smears showed enlarged granules within polymorphonuclear leukocytes (PMNs) and monocytes (Fig. 2e), a diagnostic feature of CHS. Immunofluorescence with the lysosomal marker Lamp1 in MEFs from ΔLYST-B6 mice depicted enlarged, perinuclear lysosomes, characteristic of CHS (Fig. 2f).

Significant motor behavioral differences were detected in ΔLYST-B6 mice starting at 6 months (Fig. 3a, $P = 3.09 \times 10^{-2}$). Neurological symptoms, such as a stiff tail and wobbling on hind legs appeared at 12 months. ΔLYST-B6 mice scored higher on a composite ataxia phenotype score, assessing coordination, motor impairment, kyphosis, and gait, compared to WT mice from 6 to 18 months (Fig. 3a, $P \leq 3.09 \times 10^{-2}$).

Balance and coordination were further investigated using the vertical (inverted) pole test, which assesses basal ganglia-related movement disorders[14–16]. ΔLYST-B6 mice had a longer turnaround time at 9, 12, and 18 months (Supplementary Fig. 2a, $P \leq 1.43 \times 10^{-23}$) and significantly longer descent time at 12 months (Fig. 3b, $P = 0.01$). ΔLYST-B6 exhibited difficulties descending directly, often sliding or corkscrewing down the pole, unlike WT mice.

Motor balance was also evaluated using the wide and narrow beam tests. ΔLYST-B6 mice took longer to cross both the 12-mm (Fig. 3c, $P \leq 3.36 \times 10^{-2}$) and 24-mm (Fig. 3c, $P \leq 2.90 \times 10^{-2}$) beam at 12 and 18 months. They also had more foot slips than WT on the 24-mm beam at 9-months (Supplementary Fig. 2b, $P = 3.25 \times 10^{-2}$) and on the 12-mm beam at 8, 12, and 18 months (Supplementary Fig. 2b, $P \leq 1.97 \times 10^{-2}$).

Gait analysis though inkblot footprint tests (Fig. 3d) showed significantly shorter stride lengths (Fig. 3e, $P \leq 2.11 \times 10^{-4}$) and out-toeing (Fig. 3e, $P \leq 4.82 \times 10^{-6}$) in all four paws of ΔLYST-B6 mice compared to WT. No significant differences were observed in paw stance width or stride angle (Fig. 3e, $P \geq 0.585$ and $P = 0.676$, respectively).

### Central and peripheral nervous system involvement in the ΔLYST-B6 mouse model

Histological and immmunohistological survey of the whole brain did not show gross morphological abnormalities in the ΔLYST-B6 mice (Supplementary Fig. 3a). Upon closer inspection, however, we observed notable alterations in the Purkinje cell layer of the cerebellum (Supplementary

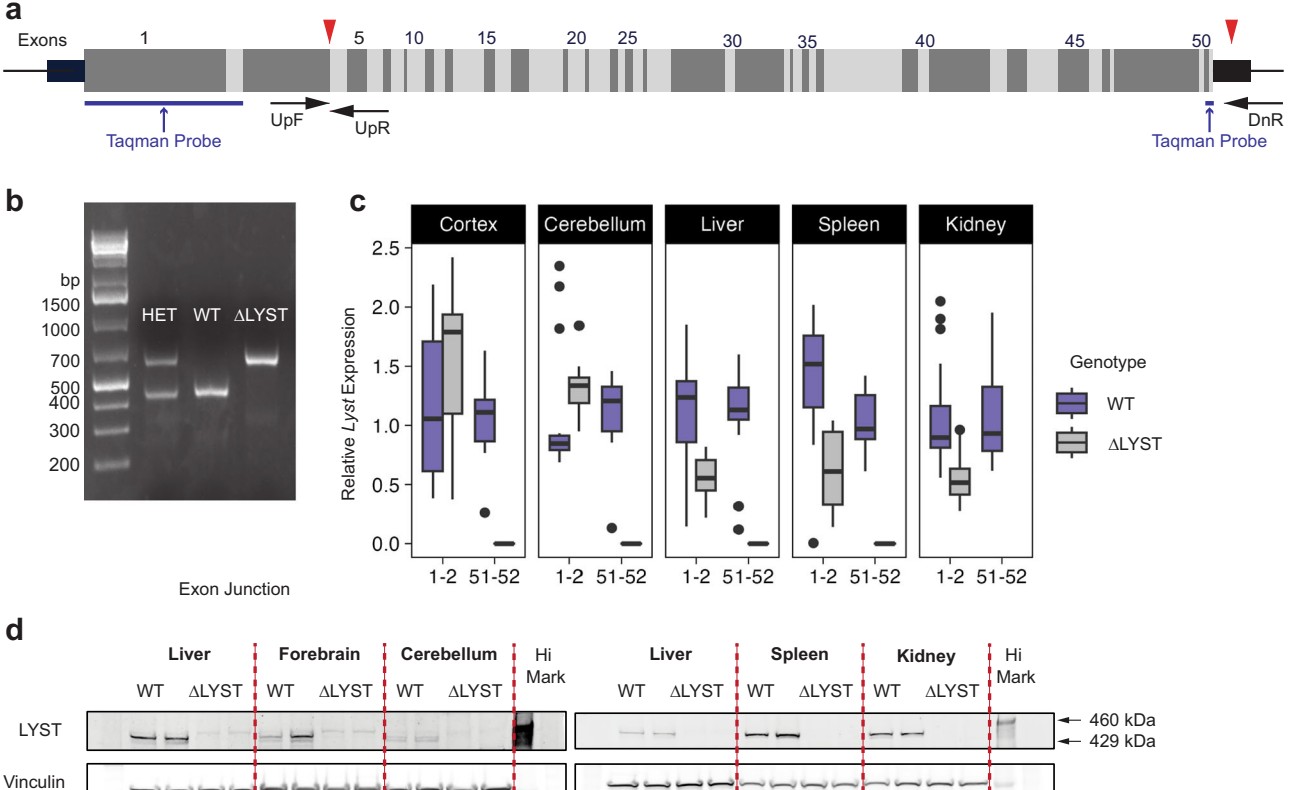

**Fig. 1 | Generation and molecular validation of a mouse model lacking *Lyst*.**
**a** Diagram of exons 1-53 of murine *Lyst* and location of the two sgRNA target sites for CRISPR/Cas9 gene editing at exon 4 and exon 53 (red arrowheads). The thick left black bar represents the 5' UTR, and the thick right black bar represents the 3' UTR. Target sites of the genotyping primers are represented by black arrows. Locations of TaqMan probes spanning exon-exon boundaries 1-2 and 52-53 for qPCR analysis are represented by blue bars. **b** Representative image of a 1.5% agarose gel showing PCR genotyping fragment sizes for heterozygous, WT, and ΔLYST-B6 mice. The small 447-bp band represents the WT allele, while the larger 642-bp band represents

the mutant allele. **c** Relative *Lyst* mRNA expression quantified by qPCR in the forebrain, cerebellum, liver, spleen, and kidney by targeting exons 1-2, before the CRISPR cut site, and 52-53, after the CRISPR cut site (*n* = 2). Student's two-sample t-test: *$P < 0.05$, **$P < 0.01$, ***$P < 0.001$. **d** LYST protein levels detected by western blot in forebrain, cerebellum, liver, spleen, and kidney of WT and ΔLYST-B6 mice (*n* = 2). Vinculin was used as a loading control. DnR downstream reverse primer, UpF upstream forward primer, UpR upstream reverse primer, Het heterozygous, WT wild-type, ΔLYST *Lyst* homozygous knockout (C57BL/6J background).

Fig. 3b). Immunohistochemical analysis with Calbindin-28K indicated large areas of Purkinje cell loss in 12-month-old ΔLYST-B6 mice; with extensive loss by 18 months (Fig. 4a). Quantification of Purkinje cells per mm revealed decreased median Purkinje cell density in ΔLYST-B6 at all age points. However, loss was only statistically significant at 18 months (Fig. 4b, $P = 4.17 \times 10^{-7}$).

Examination of lysosomal patterns in Purkinje cells and surrounding glial cells revealed no enlarged lysosomes in Purkinje cells of the ΔLYST-B6 mice. However, enlarged perinuclear lysosomes were seen in GFAP-positive cells, likely Bergmann glia, in the Purkinje cell layer,

Peripheral neuropathy was evident in ΔLYST-B6 mice, with uneven myelin decompaction of sciatic nerves at 3 months, worsening with age. By 24 months, sciatic nerves showed further degradation, apoptotic whirling, and cellular debris accumulation (Fig. 4d). Significant loss of peripheral nerve fibers was observed at 3 and 24 months ΔLYST-B6 mice (Fig. 4e, $P \leq 0.01$) when compared to WT mice.

## Multi-omics analysis in brain regions revealed altered lipidomic and transcriptomic profiles in ΔLYST-B6 mice
Lipidomic analysis of 18-month-old mice, which can infer lysosomal metabolism[17], showed distinct alterations in lipid profiles, especially in the cerebellum. Partial least squares discriminant analysis (PLS-DA)[18] showed clear segregation of genotypes across all ages and brain regions (Fig. 5a, b). Hierarchical clustering of significantly altered lipids revealed no differences in positive ion lipid level analyses (Fig. 5c, d), but with clear segregation of

genotypes in negative ion lipid level analyses in forebrain and cerebellum (Fig. 5e, f). In the forebrain, only negative ion lipids in the ceramide (Cer) class were consistently altered within each genotype (Fig. 5e). The significantly altered negative ion lipids in the cerebellum of ΔLYST-B6 (Fig. 5f, Supplementary Data 4a and 4b, $P \leq 2.75 \times 10^{-3}$) included phosphatidylserines (PS), ceramides (Cer), biotinylated phosphatidylethanolamine (BiotynlPE), lysophosphatidylglycerol (LPG), phosphatidylcholines (PC), and phosphatidylinositol (PI) (Fig. 5f). Decreased negative ion lipids were limited to the monohexosylceramides (Hex1Cer) and dihexosylceramides (Hex2Cer) classes (Fig. 5f). The top five differential lipid classes can be found in Table 2 and the top five differential individual lipids can be found in Table 3.

Transcriptomic analysis of LYST-deficient brains revealed significant enrichment of differentially expressed genes (DEGs) involved in immunological pathways, such as immune effector processes, microglia phagocytosis, inflammatory response, and synapse pruning. Principal component analysis (PCA) showed poor segregation of genotypes in 3- and 18-month-old forebrain (Fig. 6a, b) and 3-month-old cerebellum tissues (Fig. 6c), but clear segregation in the 18-month cerebellum (Fig. 6d). DEGs (Table 4, Supplementary Data 2) in the forebrain of ΔLYST-B6 mice revealed 79 DEGs at 3 months (Fig. 6e) and 47 DEGs at 18 months (Fig. 6f) compared to WT. In the cerebellum, there were 14 DEGs at 3 months (Fig. 6g) and 198 DEGs at 18 months of age (Fig. 6h). These comparisons demonstrate that LYST deficiency is associated with broad, progressive signaling changes in the central nervous system, particularly in the cerebellum.

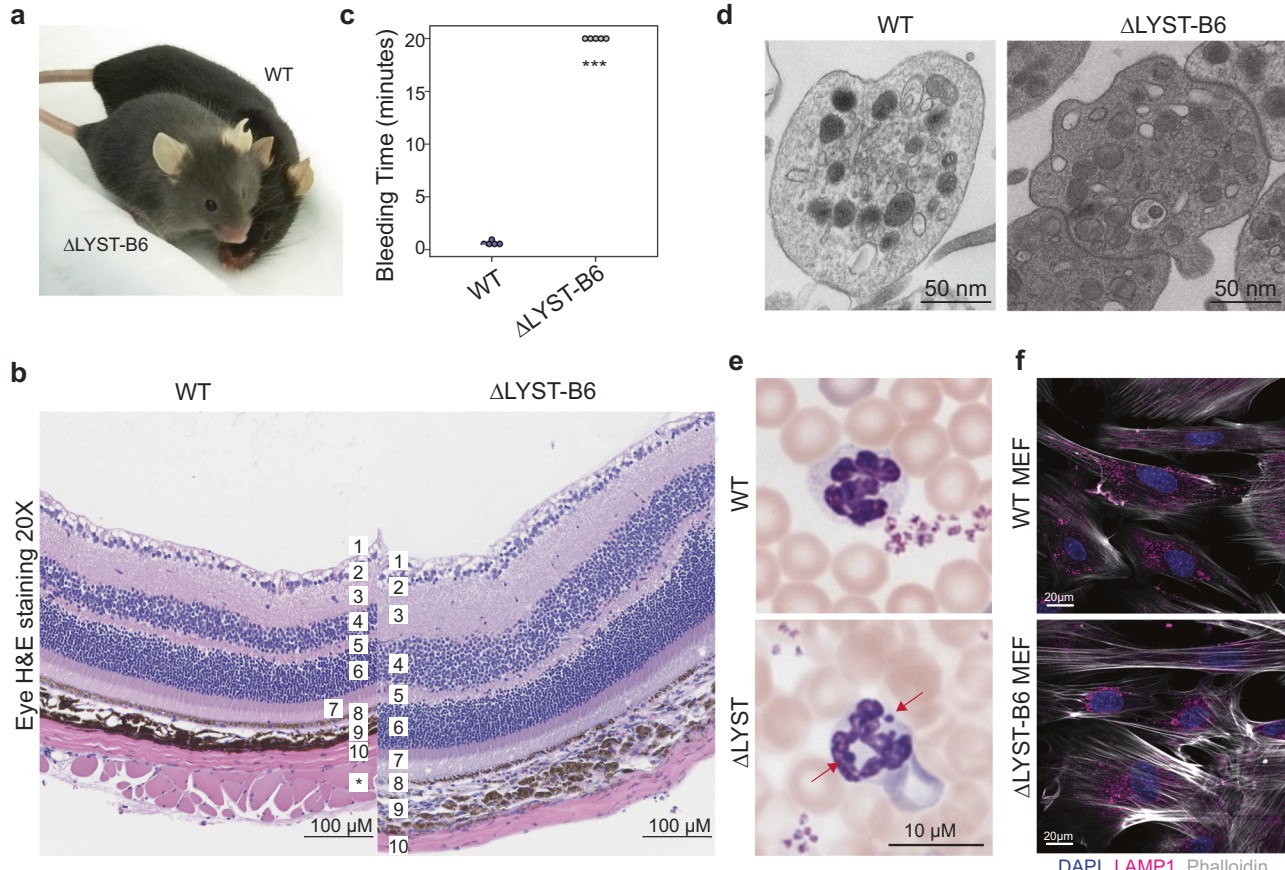

**Fig. 2 | Phenotypic characterization of the ΔLYST-B6 mouse model. a** ΔLYST-B6 mouse next to a WT control showing a lighter coat and skin color. **b** Representative H&E staining of the retina showing decreased pigmentation in the retinal pigmented epithelium in ΔLYST-B6 mice. 1. Nerve, 2. Ganglion cells, 3. Inner plexiform layer, 4. Inner nuclear layer, 5. Outer plexiform layer, 6. Outer nuclear layer, 7. Rods and cones, 8. Retinal Pigmented epithelium, 9. Choroid, 10. Sclera. In WT, * refers to an additional layer of muscle, which is oriented cross-sectionally. **c** Bleeding assay was performed on three-month old male mice under anesthesia by tail tip transection, immersing the tail in saline at 37 °C, continuously monitoring bleeding patterns and timing the bleed duration. All mutant mice ($n = 5$) had a bleeding defect when compared to wild-type mice ($n = 5$). Student's two-sample t-test: ***$P < 0.001$. **d** Whole-mount electron microscopy image of WT and ΔLYST-B6 platelets. A reduction of platelet dense granules can be seen in the ΔLYST-B6 platelet image. **e** Representative neutrophil in WT and ΔLYST-B6 peripheral blood smear. Large perinuclear granules (arrows) seen in ΔLYST-B6 cells resemble those seen in CHS patients. **f** Immunofluorescent staining of LAMP1 (lysosomes) and fluorescent staining using phalloidin (actin/cell boundaries) and DAPI (nuclei) of WT and ΔLYST-B6 mouse embryonic fibroblasts. ΔLYST-B6 mice exhibit the characteristic feature of enlarged lysosomes clustered around the nucleus, confirming the CHS phenotype. WT, wild-type; ΔLYST-B6, *Lyst* homozygous knockout (C57BL/6J background).

Hierarchical clustering of DEGs showed clear segregation of WT and ΔLYST-B6 samples across all ages and brain areas, with consistent patterns within genotypes in the cerebellum (Fig. 6i–l). Gene Ontology (GO) analysis of DEGs highlighted enriched immune pathways in the cerebellum of 18-month-old ΔLYST-B6, including neutrophil degranulation, microglia-mediated phagocytosis, and complement activation of neuroinflammation (Table 5, $P \le 5.44 \times 10^{-3}$). Similar changes were observed in the forebrain at 18 months, although with inconsistent Z-scores within genotypes (Supplementary Data 3) (Fig. 6j).

In summary, our omics results reveal more alterations in RNA expression and lipid levels in the cerebellum of ΔLYST-B6 mice compared to the forebrain. These data suggest that the neurological disease process of CHS is driven by changes in the cerebellum.

## Discussion

The ΔLYST-B6 mouse model offers a significant advancement in our understanding of CHS by providing a robust platform that recapitulates key features of the human disease, including progressive neurological deterioration. This model exhibits many aspects of CHS, including partial oculocutaneous albinism, prolonged bleeding time due to decreased platelet delta granules, enlarged intracellular granules in PMN leukocytes, and enlarged MEF lysosomes consistent with documented findings in cells from CHS patients and other CHS mouse models[10,13].

One of the most notable findings in the ΔLYST-B6 mice is the early onset of progressive neurological symptoms, which appear as soon as six months of age (Fig. 3a and Table 1), when compared to previous studies[12] (Table 1). The deletion of the majority of the LYST gene and resulting absence of functional protein suggests a correlation between the severity of genotype and the severity of phenotype in the ΔLYST-B6 compared to other LYST mouse models. This supports the ΔLYST-B6 mouse as a robust model for future therapeutic efforts, especially those focusing on modifying the neurologic sequelae of disease.

Impaired motor performance in previous CHS mouse models has been associated with Purkinje cell degradation in the cerebellum and axonal deterioration in peripheral nerves[9]. In our study, while the Purkinje cell loss is not statistically significant until 18-months (Fig. 4b, $P = 4.17 \times 10^{-7}$), the downward trend in median Purkinje cell counts in ΔLYST-B6 mice from 3 to 18 months suggests that Purkinje cell loss is progressive, contributing to poor motor coordination and ataxia. In ΔLYST-B6 mice, the enlarged lysosomes, characteristic in other cell types in CHS and in neurons in culture[19], were located in GFAP-positive glial cells surrounding the Purkinje cells that are presumed to be Bergmann

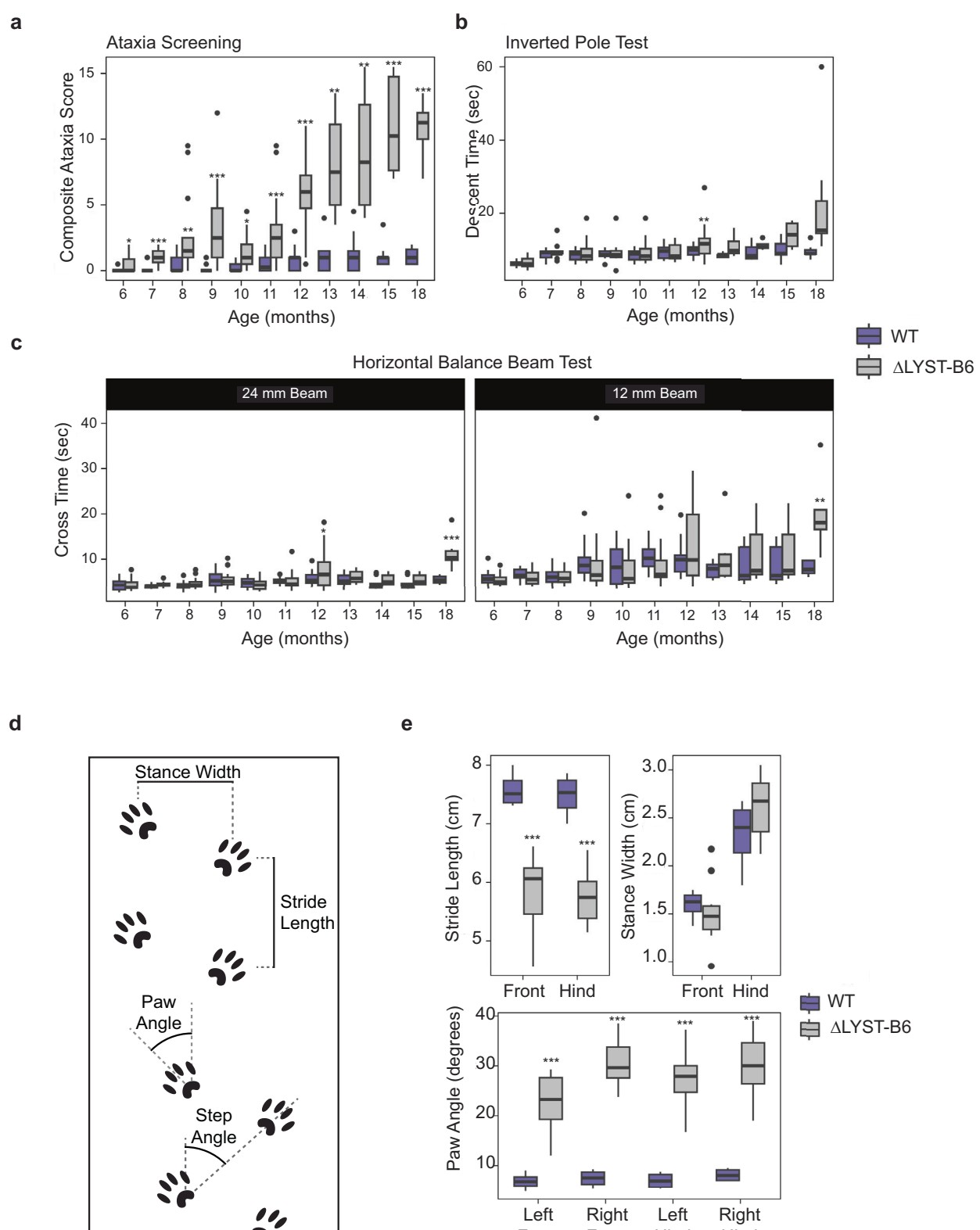

glia. These cells are critical for the maintenance and growth of cerebellar Purkinje cells[20–23], and are implicated in the development of disease[24–26] and loss of Purkinje cells[27].

A second contributor to the motor impairment in ΔLYST-B6 mice is the progressive loss of axons in the peripheral sciatic nerves of 3- and 24-month-old ΔLYST-B6 mice (Fig. 4e, $P \leq 0.01$), suggesting that mutant mice have peripheral neuropathy similar to that of CHS patients. Similar whole-mount EM findings were reported in other CHS models[12], albeit at a later age compared to ΔLYST-B6 mice (Table 1). The earlier onset and tracked progression of these findings confirms that ΔLYST-B6 mice recapitulate CHS disease and expands upon our understanding of the progressive, length-dependent neuropathy of CHS patients.

Fig. 3 | Neurobehavioral analyses of the ΔLYST-B6 mouse model. a Average cumulative composite ataxia scores from ataxia screening of WT and ΔLYST-B6 mice from the age of 6 to 18 months. 6 mo: $n = 18$ ΔLYST-B6 (7 male, 11 female), n = 17 WT (9 male, 8 female); 7 mo: $n = 18$ ΔLYST-B6 (7 male, 11 female), $n = 17$ WT (9 male, 8 female); 8 mo: $n = 17$ ΔLYST-B6 (6 male, 11 female), $n = 16$ WT (9 male, 7 female); 9 mo: $n = 26$ ΔLYST-B6 (9 male, 17 female), $n = 18$ WT (5 male, 13 female); 10 mo: $n = 11$ ΔLYST-B6 (female only), $n = 7$ WT (female only); 11 mo: $n = 17$ ΔLYST-B6 (6 male, 11 female), $n = 16$ WT (9 male, 7 female); 12 mo: $n = 31$ ΔLYST-B6 (11 male, 20 female), $n = 31$ WT (17 male, 14 female), Heterozygous $n = 3$ (1 male, 2 female); 13-15 mo: $n = 6$ ΔLYST-B6 (male only), $n = 9$ WT (male only); 18 mo: $n = 7$ ΔLYST-B6 (3 male, 4 female), $n = 7$ WT (female only). b Average descent times from the inverted pole test of WT and ΔLYST-B6 mice from the age of 6 to 18 months. 6 mo: $n = 18$ ΔLYST-B6 (7 male, 11 female), $n = 19$ WT (9 male, 10 female); 7 mo: $n = 18$ ΔLYST-B6 (7 male, 11 female), $n = 19$ WT (9 male, 810 female); 8 mo: $n = 17$ ΔLYST-B6 (6 male, 11 female), $n = 19$ WT (9 male, 10 female); 9 mo: $n = 14$ ΔLYST-B6 (9 male, 5 female), $n = 11$ WT (5 male, 6 female); 10 mo: $n = 17$ ΔLYST-B6 (6 male, 11 female), $n = 19$ WT (9 male, 10 female); 11 mo: $n = 17$ ΔLYST-B6 (6 male, 11 female), $n = 18$ WT (9 male, 9 female); 12 mo: $n = 31$ ΔLYST-B6 (11 male, 20 female), $n = 32$ WT (17 male, 15 female), $n = 3$ Heterozygous (1

male, 2 female); 13 mo: $n = 6$ ΔLYST-B6 (male only), $n = 9$ WT (male only); 14-15 mo: $n = 4$ ΔLYST-B6 (male only), $n = 9$ WT (male only); 18 mo: $n = 7$ ΔLYST-B6 (3 male, 4 female), $n = 7$ WT (female only). c Average cross time of 24-mm beam and 12-mm beam from the horizontal balance beam test of WT and ΔLYST-B6 mice from the age of 6 to 18 months. 6 mo: $n = 18$ ΔLYST-B6 (7 male, 11 female), $n = 18$ WT (9 male, 9 female); 7 mo: $n = 18$ ΔLYST-B6 (7 male, 11 female), $n = 18$ WT (9 male, 9 female); 8 mo: $n = 18$ ΔLYST-B6 (7 male, 11 female), $n = 18$ WT (9 male, 9 female); 9 mo: $n = 25$ ΔLYST-B6 (9 male, 16 female), $n = 29$ WT (14 male, 15 female); 10 mo: $n = 17$ ΔLYST-B6 (6 male, 11 female), $n = 17$ WT (9 male, 8 female); 11 mo: $n = 17$ ΔLYST-B6 (6 male, 11 female), $n = 17$ WT (9 male, 8 female); 12 mo: $n = 31$ ΔLYST-B6 (11 male, 20 female), $n = 31$ WT (17 male, 14 female), $n = 3$ Heterozygous (1 male, 2 female); 13-15 mo: $n = 6$ ΔLYST-B6 (male only), $n = 9$ WT (male only); 18 mo: $n = 7$ ΔLYST-B6 (3 male, 4 female), $n = 7$ WT (female only). Inkblot gait analysis measurements (d) and average stride length, stance width, and paw angle (e) from inkblot gait analysis of WT and ΔLYST-B6 mice at the age of 12 months. $n = 10$ ΔLYST-B6 (3 male, 7 female), n = 4 WT (2 male, 2, female). Student's two-sample $t$-test: *$P < 0.05$, **$P < 0.01$, ***$P < 0.001$. WT, wild-type; ΔLYST-B6, *Lyst* homozygous knockout (C57BL/6 J background).

## ΔLYST-B6 show regional alteration in brain lipidomics and transcriptomics

Elevated levels of pro-inflammatory lipids (PS) and reduced levels of anti-inflammatory lipids (OAHFA) (Fig. 5 and Table 4) suggest a heightened neuroinflammatory environment in the cerebellum of ΔLYST-B6 mice. PS facilitates complement-mediated synaptic pruning by microglia receptor TREM2[28,29], a protein implicated in the regulation of autophagy in microglia. This interplay of lipid and microglial autophagy has been identified in several neurodegenerative pathologies such as AD and PD[29–32]. The pro-inflammatory effects of PS seem to be consistent with reduced levels of OAHFA, implicated to have anti-inflammatory activities[33,34], and reduced in brain tissue of frontotemporal dementia patients[35]. It should be noted that the inflammatory contribution (anti- or pro-) of PS may depend on its localization to the inner or outer leaflet of the plasma membrane[28,29], which our study did not assess.

These lipidomics findings are consistent with the observed upregulation of immune pathways in our transcriptomic analyses, indicating that neuroinflammation plays a significant role in the neurological deterioration seen in CHS. The increased expression of microglia-associated genes and complement cascade genes in the cerebellum of ΔLYST-B6 mice further supports the involvement of neuroinflammatory processes in CHS pathology. Two genes in these pathways upregulated in the cerebellum of 18-month-old ΔLYST-B6 mice, *TYROBP* and *TREM2*, are both implicated in microglial activation in AD. This is ultimately associated with C1q activation[27], which results in an inflammatory response that triggers further microglia recruitement[27]. The patterns of upregulated immunological activation in known microglia-associated genes could implicate microglial dysregulation in CHS pathology. This is further supported by the increased expression of the complement cascade genes *C1qa*, *C1qb*, *C1qc*, *C3*, *C4b*, and the complement receptor encoded by *C3ar1* identified by RNA-seq. Interestingly, GO analysis in forebrain and cerebellum at 3 months of age did not indicate similar changes in immune pathways, suggesting that immune signaling is not widely upregulated in the brain of ΔLYST-B6 mice until there is behavioral evidence of neurodegeneration. Further studies evaluating the onset and development of neuroinflammation will be required to identify the role of glial pathophysiology in CHS.

## Contribution of ΔLYST genetic background in phenotype severity

Due to previously reported contributions of genetic background to the development of disease and Purkinje cell loss[10], we also crossed our ΔLYST-B6 to the DBA/2J (ΔLYST-DBA; official name: DBA/2J-*Lyst*[em1Mldn/Mldn]; MGI 7797858) to analyze motor function and brain histology. Motor behavioral differences were detected in ΔLYST-DBA mice by cumulative ataxia screening starting at 3 months of age and became significant at 4 months of age, continuing to widen until 7 months of age (Supplementary

Fig. 7a, $P \leq 5.96 \times 10^{-2}$). Additionally, ΔLYST-DBA took significantly longer to cross the 12 mm beam at 5 months of age (Supplementary Fig. 7b, $P = 4.42 \times 10^{-3}$) and the 24 mm beam at 4 months of age (Supplementary Fig. 7b, $P = 1.59 \times 10^{-2}$) than the WT controls.

Immunofluorescent analysis of Calbindin d-28k revealed large areas of Purkinje cell loss in 3-month-old ΔLYST-DBA mice compared to WT mice. This trend continued to 4- and 7-month-old mice but was not statistically significant at these age points (Supplementary Fig. 8, $P \geq 0.13$).

## Limitations of the study

While this study provides valuable clues towards the neurological pathophysiology in CHS, it has important limitations.

Our results revealed interesting histological changes in the cerebellum, specifically, the Purkinje cells, but additional studies are required to confirm whether the Purkinje cells are the primary cells affected in CHS or are secondarily affected by other upstream processes. The observation that the enlarged lysosomes in Bergmann glia are noted prior to the onset of measurable motor incoordination, could suggest a mechanism of Purkinje cell loss whereby dysfunctional Bergmann glia result in deteriorating Purkinje cell health and eventual death. The plausible role of Bergmann glia in the pathology and progression of CHS should be a focus of future studies.

Additionally, our -omics findings are limited in their spatial granularity. Upregulated immune findings from bulk RNA-seq could represent either diffuse neuroinflammation or the concentration of immune signaling near degenerative cell populations. Similarly, untargeted lipidomic analysis from bulk tissue is unable to account for subcellular localization, which may clarify their pro- or anti-inflammatory effects. Future work disentangling the spatial distribution of transcriptomic and lipidomic alterations seen in the brains of ΔLYST-B6 mice will provide more precise information on the pathophysiology of CHS.

Finally, while we were able to show that the ΔLYST-DBA manifested with an earlier phenotype compared to ΔLYST-B6, our studies have not identified the precise set of genes responsible that could account for this phenotypic variation. Further research is needed to identify the specific genetic factors that contribute to this phenotypic variation, which could provide valuable insights into the mechanisms underlying CHS.

## Conclusion

The ΔLYST-B6 mouse model represents a significant advancement in the study of CHS, providing a comprehensive platform to investigate the pathophysiology of the disease and to explore potential therapeutic interventions. The early onset and progressive nature of neurological symptoms in this model make it particularly valuable for studying the mechanisms of neurodegeneration could pave the way for personalized therapeutic approaches for CHS patients.

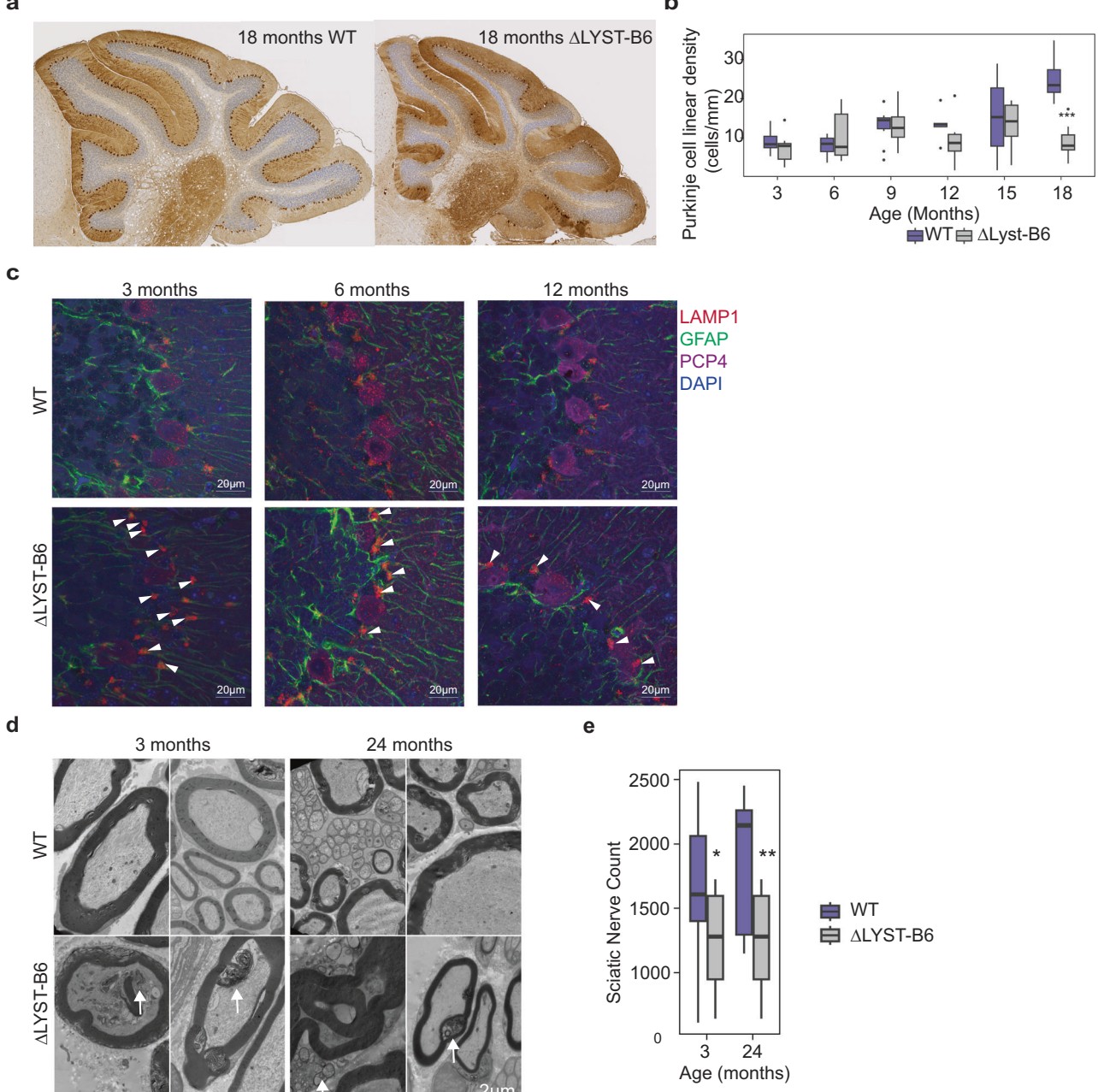

**Fig. 4 | Histological, TEM, and immunofluorescent analyses of the ΔLYST-B6 forebrain, cerebellum and sciatic nerve. a** Representative image of Calbindin-28k stained cerebellum of 18-month-old WT control and ΔLYST-B6. Purkinje cells are dark brown cells above the blue-grey granular layer. ΔLYST-B6 mice showed a large loss of the Purkinje cell layer. **b** Average linear density of Purkinje cells counted in the cerebellum per millimeter (mm) of 3-, 6-, 9-, 12-, 15-, 18-month-old mice. 3 mo: $n = 10$ ΔLYST-B6 (5 male, 5 female), $n = 10$ WT (5 male, 5 female); 6 mo: $n = 9$ ΔLYST-B6 (4 male, 5 female), $n = 10$ WT (5 male, 5 female); 9 mo: $n = 9$ ΔLYST-B6 (4 male, 5 female), n = 9 WT (4 male, 5 female), $n = 1$ Heterozygous (male); 12 mo: $n = 8$ ΔLYST-B6 (4 male, 4 female), $n = 7$ WT (5 male, 2 female), $n = 1$ Heterozygous (male); 15 mo: $n = 6$ ΔLYST-B6 (3 male, 3 female), $n = 4$ WT (female only); 18 mo: $n = 10$ ΔLYST-B6 (5 male, 5 female), $n = 10$ WT (5 male, 5 female). ANOVA

analysis: *** = $P < 0.001$. **c** Immunofluorescent staining of LAMP1 (lysosomes), GFAP (glial cells), and PCP4 (Purkinje cells), and fluorescent staining using DAPI (nuclei) of WT and ΔLYST-B6 brains at 3-, 6-, and 12-months old. Perinuclear lysosomes (arrowheads) were visible in glial cells of ΔLYST-B6 mice. **d** Representative TEM images of murine sciatic nerve of WT and ΔLYST-B6 at 3 and 24 months of age. Myelin is dark grey surrounding neurofibrils. Apoptotic whirling (arrows) in myelin layers and accumulation of cellular debris was observed in ΔLYST-B6. Scale bar, 2 μm **e.** Average number of toluidine-blue stained sciatic nerves in 3- and 24-month WT ($n = 2$) and ΔLYST-B6 ($n = 2$) mice. Student's two-sample $t$-test: *$P < 0.05$, **$P < 0.01$. WT, wild-type; ΔLYST-B6, *Lyst* homozygous knockout (C57BL/6J background).

## Materials and methods
### Animals
All experiments in mice were approved by the NIH NHGRI Animal Care and Use Committee (ACUC), under the protocol G-14-3, "Mouse models for disorders of lysosomes and lysosomal-related organelles". All animals were housed in a pathogen-free environment with food and water supplied *ad libitum* under a 12 h light/dark cycle (from 06:00 to 18:00) in an AAALAC-approved facility following National Institutes of Health (NIH) and NHGRI ACUC-approved guidelines. We have complied with all relevant ethical regulations for animal use.

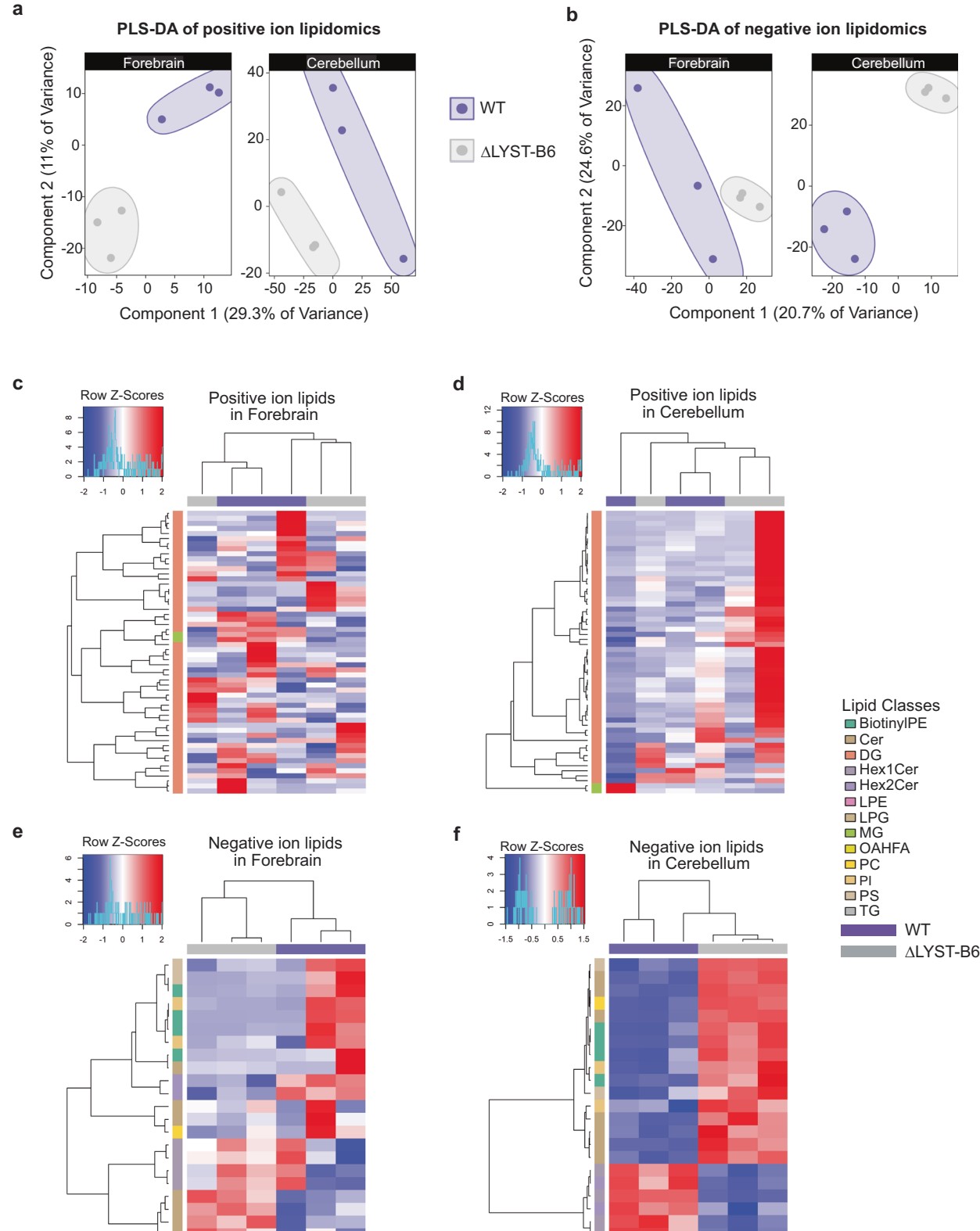

**Fig. 5 | Lipidomics analysis of 18-month-old ΔLYST-B6 cerebellum and forebrain. a** Partial least squares-discriminant analysis (PLS-DA) of variation in negative ion lipidomics in the cerebellum and forebrain of WT ($n = 3$) and ΔLYST-B6 mice ($n = 3$). **b** PLS-DA of variation in positive ion lipidomics in the cerebellum and forebrain of WT (purple) and ΔLYST-B6 (grey) mice. **c–f** Heatmaps showing expression of lipids of in the brains of ΔLYST-B6 and WT mice. Red represents of over-expression of the lipid or lipid class, while blue shows under-expression. **c** Positive ion lipids in the forebrain. **d** Positive ion lipids in the cerebellum. **e** Negative ion lipids in the forebrain. **f** Negative ion lipids in the cerebellum. WT, wild-type; ΔLYST-B6, *Lyst* homozygous knockout (C57BL/6J background).

To knockout *Lyst*, we designed two single CRISPR/Cas9 guide RNA sites targeting exons 4 and 53 (Supplementary Table 1a). Fertilized eggs were collected from superovulated 129S1/SvImJ (The Jackson Laboratory) females after mating with C57BL/6J males (The Jackson Laboratory). Cas9 protein and gRNA were delivered to the zygotes via electroporation (NEPA21 electroporator, NEPAGENE, Japan), then zygotes were surgically implanted into pseudo-pregnant recipient CB6F1 female mice. Potential founders were screened by PCR and Sanger sequencing, identifying three founders with successful deletion. A single founder with the largest deletion (Supplemental Data 1) was bred with WT C57BL/6 J (MGI:3028467) to establish the line, and was backcrossed three times to eliminate any off targets and generate germline mutants that lack *Lyst* (C57BL/6J-*Lyst* $^{em1Mldn/Mldn}$; MGI:7797890); hereon ΔLYST-B6). Knockout colony was derived from heterozygous in-crosses, and later maintained as a homozygous line from homozygous breeding. Controls were WT C57BL/6 J, originally derived from the heterozygous in-crosses and maintained in parallel with the homozygous line.

Animals were genotyped by multiplex PCR using genomic DNA extracted from a 2-mm tail trimming using the Extract-N-Amp Tissue PCR Kit (XNAT2, Sigma-Aldrich). The multiplex PCR was designed with three primers in one reaction: forward primers upstream of and within the CRISPR/Cas9 cut site, and a reverse primer downstream of the cut site (Supplementary Table 1b). PCR fragments (ΔLYST band at 642 bp and WT band at 447 bp) were resolved on a 1.5% agarose gel with a GeneRuler 1 Kb Plus DNA Ladder (SM1333, Thermo Scientific).

To generate the ΔLYST mutation on a different genetic background, the ΔLYST-B6 mice were backcrossed to DBA/2J line (MGI:2684695), purchased from Jackson Laboratory (000671, Jackson Laboratory, Bar Harbor, ME). A total of nine backcrosses were made using both recipient and donor females and their respective opposite males. Background genotyping was determined by MiniMUGA Background Analysis v0009 until we obtained DBA/2J-*Lyst* $^{em1Mldn/Mldn}$; MGI 7797858 (ΔLYST-DBA/2) (Supplementary Fig. 6, ΔLYST-DBA). Animals were genotyped by multiplex PCR using primers and methods described above (regarding multiplex PCR).

## Animal tissue collection

Animals were euthanized by intraperitoneal lethal injection (>300 mg/kg) of 1.25% tribromoethanol (Avertin). For frozen tissues, including brain, liver, kidney, and spleen, tissues were promptly harvested and frozen in liquid nitrogen and stored at −80℃ until downstream processing. For histology, tissues were placed in 4% paraformaldehyde and fixed overnight, and either processed for paraffin embedding, or immersed in 15% to 30% sucrose for OCT embedding. Sciatic peripheral nerves were collected and immediately fixed in 2% glutaraldehyde. Mouse embryonic fibroblasts (MEFs) from WT and mutant mouse strains were isolated from embryos. Briefly, pregnant dams were euthanized by intraperitoneal lethal injection of 1.25% tribromoethanol (>300 mg/kg) at 15.5 days of gestation, embryos were harvested individually from the uterus, euthanized by surgical decapitation, and tissues were minced into a slurry and plated in culture with complete Dulbecco's modified Eagle's medium supplemented by 15% fetal bovine serum, 100 IU/mL penicillin, and 100 μg/mL streptomycin.

Additionally, for ΔLYST-DBA/2, animals were euthanized by intraperitoneal lethal injection of 1.25% tribromoethanol (>300 mg/kg) and then perfused (transcardiac perfusion) with PBS to wash the tissues. Tissues were harvested and then immediately placed in 4% paraformaldehyde and fixed overnight before being transferred to 70% ethanol for paraffin embedding.

## Quantitative PCR analysis

Tissues were homogenized with TRIzol Reagent (15596028, Invitrogen) using zirconium beads (D1032-15, Benchmark Scientific) and the BeadBug microtube homogenizer (D1030-E, Benchmark Scientific). Total RNA was extracted using the miRNeasy Mini Kit (217004, QIAGEN) following the manufacturer's directions, followed by a DNase digestion using the DNAse-free DNA Removal Kit (AM1906, Invitrogen) and quantified using the NanoDrop 1000 Spectrophotometer. Reverse transcription was performed using the High-Capacity RNA-to-cDNA Kit (4387406, Applied Biosystems) according to the manufacturer's instructions.

Quantitative PCR was performed using the Quant Studio 6 Pro Real-Time PCR System (Applied Biosystems). Pre-designed TaqMan Gene

**Table 2 | Top 5 differentially-expressed lipid classes in the forebrain and cerebellum of 18-month-old WT and ΔLYST-B6 mice**

| Brain Region | Lipid Class | Ion | Log$_2$(Fold Change) | *P*-value |
|---|---|---|---|---|
| Forebrain | BiotinylPE | Negative | −2.30 | 6.87E−5 |
| Forebrain | Hex1Cer | Negative | −1.09 | 1.46E−4 |
| Forebrain | LPE | Negative | 0.59 | 6.31E−4 |
| Forebrain | LPG | Negative | 1.24 | 8.12E−4 |
| Forebrain | Hex2Cer | Negative | −1.38 | 1.60E−3 |
| Cerebellum | TG | Positive | 1.20 | 3.78E−14 |
| Cerebellum | BiotinylPE | Negative | 2.30 | 3.50E−7 |
| Cerebellum | OAHFA | Negative | −1.84 | 6.30E−6 |
| Cerebellum | PS | Negative | 0.65 | 3.09E−4 |
| Cerebellum | Hex1Cer | Positive | −0.66 | 7.26E−4 |

*BiotinylPE* Biotinylated phosphatidylethanolamine, *Hex1Cer* hexosylceramide, *LPE* lysophosphatidylethanolamine, *LPG* lysophosphatidyl-glycerol, *Hex2Cer* dihexosylceramide, *TG* triglyceride, *OAHFA* (O-acyl)-ω- hydroxy fatty acid, *PS* phosphatidylserine, MG monoradylglycerolipid, Cer ceramide, DG diacylglycerols.

**Table 3 | Top 5 differentially-expressed individual lipid levels in the forebrain and cerebellum of 18-month-old WT and ΔLYST-B6 mice**

| Brain Region | Lipid | Ion | Log$_2$(FoldChange) | *P*-value |
|---|---|---|---|---|
| Forebrain | MG(20:1) + NH4 | Positive | −0.71 | 5.37E−6 |
| Forebrain | Hex2Cer(d38:1)-H | Negative | −0.63 | 2.00E−5 |
| Forebrain | Cer(d40:0 + O)-H | Negative | 1.09 | 3.90E−5 |
| Forebrain | MG(20:1)+Na | Positive | −0.74 | 2.45E−4 |
| Forebrain | PS(40:7) + H | Positive | 2.84 | 2.65E−4 |
| Cerebellum | DG(28:1e)+NH4 | Positive | 1.51 | 1.90E−9 |
| Cerebellum | BiotinylPE(34:6)-H | Negative | 3.53 | 5.89E−8 |
| Cerebellum | Cer(t36:1)-H | Negative | 2.30 | 6.37E−8 |
| Cerebellum | Hex1Cer(d44:2)-H | Negative | −1.11 | 4.27E−6 |
| Cerebellum | PS(42:8)-H | Negative | 1.87 | 4.41E−6 |

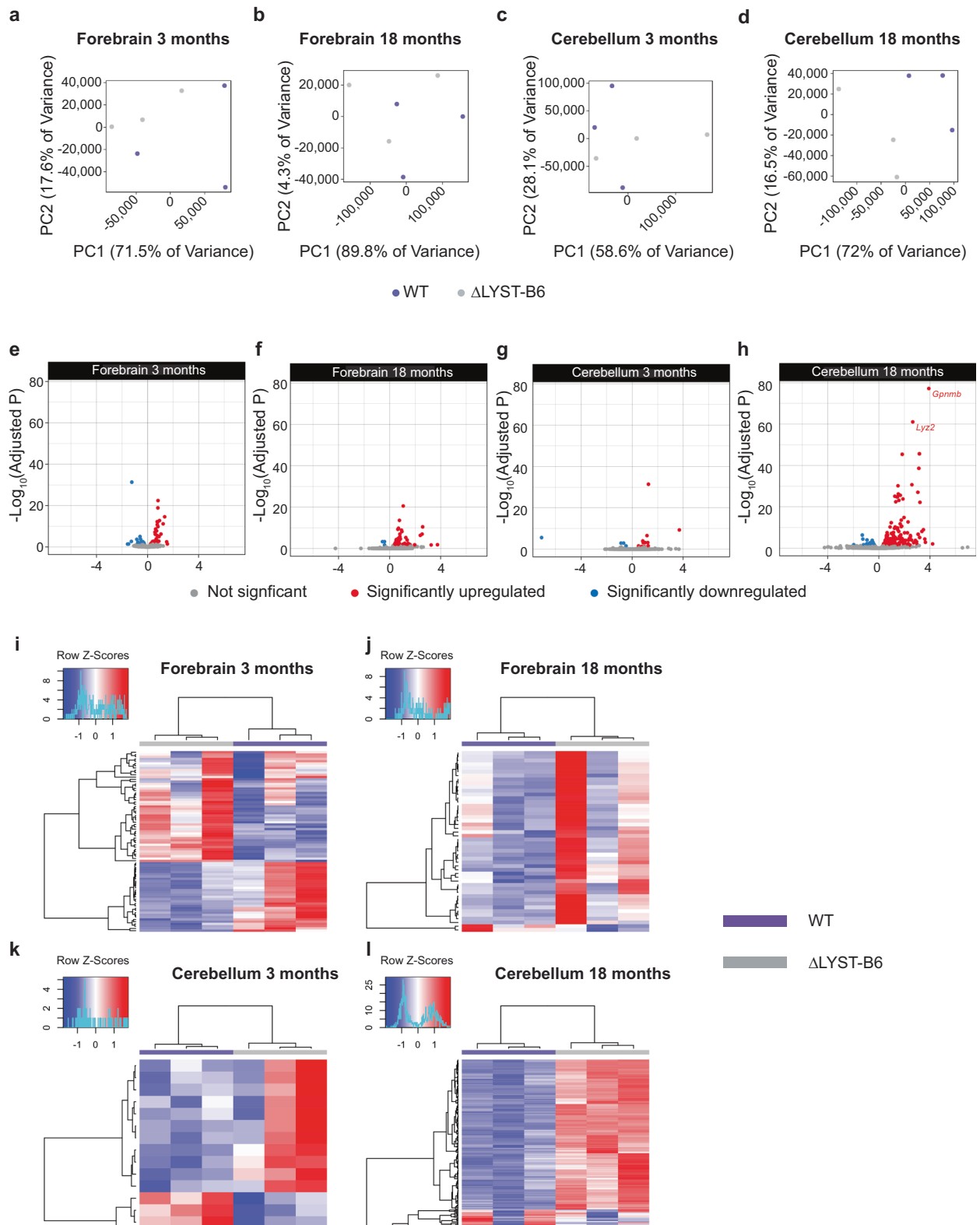

**Fig. 6 | RNA sequencing of 3- and 18-month-old ΔLYST-B6 cerebellum and forebrain.** Principal component analysis (PCA) of variation in 3-month-old forebrain (**a**), 18-month-old forebrain (**b**), 3-month-old cerebellum (**c**), and 18-month-old cerebellum (**d**). WT (*n* = 3) and ΔLYST-B6 mice (n = 3). Volcano plots showing differential expression of genes in 3- and 18-month-old cerebellum and forebrain of ΔLYST-B6 mice. Red represents significantly upregulated genes, while blue shows significantly downregulated genes. Fold change is calculated as ΔLYST-B6/WT. This analysis was performed in 3-month-old forebrain (**e**), 18-month-old forebrain (**f**), 3-month-old cerebellum (**g**), and 18-month-old cerebellum (**h**). Heatmaps showing differential expression of genes in the forebrain and cerebellum of WT (purple) and ΔLYST-B6 (grey) mice at 3- and 18-months old. Red represents significantly upregulated genes and blue represents significantly downregulated genes. This analysis was performed in 3-month-old forebrain (**i**), 18-month-old forebrain (**j**), 3-month-old cerebellum (**k**), and 18-month-old cerebellum (**l**). WT wild-type, ΔLYST-B6, *Lyst* homozygous knockout (C57BL/6 J background).

Expression Assays (Applied Biosystems) targeting the *Lyst* exon 1-2 boundary (Mm00464952_m1), the *Lyst* exon 52-53 boundary (Mm00465002_m1), and the endogenous control gene *Actb* (Mm00607939_s1) were used according to the manufacturer's specifications.

### Table 4 | Top 5 differentially-expressed genes from RNA-seq analysis of the forebrain and cerebellum in 3- and 18-month-old WT and ΔLYST-B6 mice

| Brain Region | Age (Months) | Base Mean | Log₂(FoldChange) | Adjusted P | Gene Name |
|---|---|---|---|---|---|
| Forebrain | 3 | 1246 | −1.23 | 4.79E−32 | *Btaf1* |
| Forebrain | 3 | 2044 | 0.819 | 3.96E−23 | *Pik3c3* |
| Forebrain | 3 | 2628 | 0.806 | 1.37E−19 | *Junb* |
| Forebrain | 3 | 594.4 | 1.34 | 2.65E−15 | *Lmbr1* |
| Forebrain | 3 | 967.5 | 0.936 | 1.78E−13 | *Fos* |
| Cerebellum | 3 | 2349 | 1.28 | 4.10E−32 | *Pik3c3* |
| Cerebellum | 3 | 586.4 | 1.16 | 3.01E−07 | *Lmbr1* |
| Cerebellum | 3 | 98.51 | -7.09 | 2.69E−06 | *Kpna7* |
| Cerebellum | 3 | 803.4 | 0.833 | 1.06E−4 | *Cd59b* |
| Cerebellum | 3 | 195.7 | 1.02 | 6.40E−4 | *Npas4* |
| Forebrain | 18 | 10830 | 1.07 | 2.78E−21 | *Gfap* |
| Forebrain | 18 | 1347 | 0.770 | 2.69E−14 | *H2-K1* |
| Forebrain | 18 | 78.30 | 2.59 | 3.62E−11 | *Clec7a* |
| Forebrain | 18 | 1288 | 0.695 | 1.70E−10 | *Mpeg1* |
| Forebrain | 18 | 2177 | 0.645 | 5.29E−10 | *H2-D1* |
| Cerebellum | 18 | 754.5 | 3.91 | 6.12E−78 | *Gpnmb* |
| Cerebellum | 18 | 1536 | 2.66 | 8.81E−62 | *Lyz2* |
| Cerebellum | 18 | 475.8 | 3.17 | 2.23E−46 | *C3* |
| Cerebellum | 18 | 4358 | 1.83 | 3.99E−46 | *C4b* |
| Cerebellum | 18 | 188.2 | 3.13 | 2.40E−39 | *Lilrb4a* |

Base Mean, the mean normalized gene counts across all samples; Log₂(FoldChange), Log2 of the mean normalized expression ratio (ΔLYST-B6/WT).

Each experiment included three technical replicates of each tissue sample from two biological replicates of 3-month-old ΔLYST-B6 and WT mice.

### Western blotting procedure

Tissues were homogenized with RIPA Buffer supplemented with Halt Protease Inhibitor 100X Cocktail (78430, Thermo Fisher Scientific) using bead homogenization as above. Samples were incubated on ice for 15 min and the supernatant was collected after centrifugation at 4 °C for 20 min. MEFs were homogenized using RIPA Buffer with 1X cOmplete Protease Inhibitor Cocktail (5892791001, MilliporeSigma). Protein homogenates were incubated on ice for 15 min and sonicated with the Branson Digital Sonifier 250 on ice, and then the supernatant was collected after centrifugation at 16,100 x *g* for 10 min at 4°C.

Two biological replicates of mouse tissue (40 μg total protein) and two biological replicates of MEF (70 μg total protein) lysates were solubilized and reduced using NuPAGE LDS Sample Buffer (4X) and NuPAGE Sample Reducing Agent (10X) (Invitrogen). Samples were heated at 70 °C for 5 min, centrifuged, loaded onto NuPAGE 3–8% Tris-Acetate mini gels (Invitrogen), and blotted onto nitrocellulose (iBlot2, Invitrogen). Membranes were blocked using Intercept PBS blocking buffer (LI-COR Biosciences), followed by incubation in primary antibody diluted in blocking buffer overnight at 4 °C h at room temperature. Total protein staining was performed using Revert 700 Total Protein Stain (LI-COR Biosciences). Immunodetection was performed using the Odyssey infrared fluorescence system (version 3.0, LI-COR Biosciences). Protein quantification was performed using Empiria Studio software (version 2.2.0.141, LI-COR Biosciences). Antibodies are listed in Supplementary Table 2.

### Immunofluorescence

For indirect immunofluorescence staining, MEFs at 70% confluency were fixed for 1 h at room temperature in 3% paraformaldehyde, rinsed in PBS, and incubated in blocking buffer (1% donkey serum/PBS) for 3 h at room temperature or overnight at 4 °C. Cells were then incubated with primary antibody diluted in 0.1% bovine serum albumin and 0.1% saponin in PBS for 1–2 h at room temperature or overnight at 4 °C, washed three times in 1% saponin in PBS, and incubated with the secondary antibody for 1 h at room temperature. After washing three times, cover slips were mounted using Mounting Medium with DAPI (50011, ibidi). Cells were imaged using

### Table 5 | Top 5 enriched pathways from RNA-seq analysis of the forebrain and cerebellum in 3*- and 18-month-old WT and ΔLYST-B6 mice

| Brain Region | Age (Months) | GOBP Term | Description | Enrichment Ratio | FDR |
|---|---|---|---|---|---|
| Forebrain | 3 | GO:0046683 | response to organophosphorus | 15.0 | 4.19E−5 |
| Forebrain | 3 | GO:0014074 | response to purine-containing compound | 13.2 | 5.66E−5 |
| Forebrain | 3 | GO:0042326 | negative regulation of phosphorylation | 5.15 | 1.44E−2 |
| Forebrain | 3 | GO:1901652 | response to peptide | 4.76 | 1.96E−2 |
| Forebrain | 3 | GO:0009612 | response to mechanical stimulus | 7.09 | 3.11E−2 |
| Forebrain | 18 | GO:0072376 | protein activation cascade | 33.1 | 1.89E−5 |
| Forebrain | 18 | GO:0002443 | leukocyte mediated immunity | 9.60 | 2.29E−5 |
| Forebrain | 18 | GO:0002250 | adaptive immune response | 7.50 | 6.03E−4 |
| Forebrain | 18 | GO:0019882 | antigen processing and presentation | 18.2 | 1.53E−3 |
| Forebrain | 18 | GO:0002253 | activation of immune response | 7.20 | 1.79E−3 |
| Cerebellum | 18 | GO:0002443 | leukocyte mediated immunity | 7.23 | 9.02E−14 |
| Cerebellum | 18 | GO:0006909 | phagocytosis | 10.1 | 1.07E−11 |
| Cerebellum | 18 | GO:0002250 | adaptive immune response | 6.06 | 1.07E−11 |
| Cerebellum | 18 | GO:0042110 | T cell activation | 5.06 | 1.59E−9 |
| Cerebellum | 18 | GO:0002697 | regulation of immune effector process | 5.50 | 4.45E−9 |

*No significantly dysregulated pathways in 3-month cerebellum; *GOBP* Gene Ontology Biological Process, *FDR* False Discovery Rate.

a confocal microscope (LSM 880, Zeiss Microscopy, Jena, Germany) at 20X and 63X magnification.

## Peripheral blood smears and bleeding assay

For peripheral blood smears, animals were euthanized as above and approximately 100 µL of blood was collected from the inferior vena cava; a drop of blood was smeared on a SuperFrost White Microscope Slide (4445, Thermo Scientific), dried, and stained with Wright-Giemsa Stain Kit (9990710, Epredia) according to the recommended specifications. Bleeding assay was performed on mice ($n = 5$), sedated by non-lethal (125–300 mg/kg), intraperitoneal injection of 1.25% tribromoethanol, by tail transection, where the transected tail tip was immersed in warm (37°C saline and continuously monitored. Mice with continuous bleeding for over 20 min were considered to have a clotting deficiency and were euthanized by lethal injection of 1.25% tribromoethanol.

## Analysis of tissue sections

4-5 µm paraffin-embedded tissue sections were deparaffinized, hydrated, and subsequently rinsed. H&E was done by Histoserv (Histoserv Incorporated, Germantown, MD) or the Fredrick National Laboratory for Cancer Research (Fredrick National Laboratory, Fredrick MD). For immunohistochemistry, slides were incubated with Background Sniper (BS966, Biocare Medical, Pacheco, CA) for 10 min, then Background Punisher (BP974, Biocare Medical) for 10 min, and rinsed with TBS-T. Slides were incubated with rabbit anti-Calbindin (Supplementary Table 2) for 1 h diluted in Van Gogh Yellow Diluent (PD902, Biocare Medical), washed twice in TBS-T rinses, incubated in MACH 2 (MALP521, Biocare Medical) for 30 min, then DAB+ (DB801, Biocare Medical) for 5 min. All slides were counterstained with CAT Hematoxylin (1:5, CATHE, Biocare Medical) for 2 min. For negative controls, rabbit IgG was used instead of the primary antibody.

For immunofluorescence, thawed OCT-embedded frozen brain sections were briefly rehydrated with PBS, permeabilized with 0.02% saponin in PBS for 10 min, washed with 1X PBS twice, and then blocked (2% donkey serum, 0.02% saponin, 0.075% glycine in PBS) for 1 h at room temperature. Sections were then incubated in 0.1% bovine serum albumin and 0.02% saponin in PBS overnight at 4 °C with the primary antibodies (Supplementary Table 2). After three washes in 0.01% saponin/PBS, sections were incubated in appropriate secondary antibodies (Supplementary Table 2) for 1 h at room temperature, washed three times in 0.01% saponin/PBS, and mounted with Vectashield antifade Mounting Medium with DAPI (H-1200, Vector Laboratories). Cells were imaged using a confocal microscope (LSM 880, Zeiss Microscopy, Jena, Germany) at 20X and 63X magnification.

For immunofluorescence of the cerebellum of the ΔLYST-DBA/2 mice, paraffin-embedded tissues were sagittally cut by Histoserv (Histoserv Incorporated, Germantown, MD) and placed on slides. Brain sections underwent deparaffinization in Hemo-De twice for 15 min each, followed by dehydration in ethanol (100% EtOH – 5 min x2, 95% EtOH – 5 min, 70% EtOH – 5 min, 50% EtOH – 5 min) before being dipped in dH$_2$O and rinsed in 1x PBS at room temperature on a shaker. Slides were then microwaved in 0.01 M sodium citrate at 100% power for 5 min, 50% power for 5 min, and 30% power for 5 min for antigen retrieval. Slides were then rinsed in prechilled 1x PBS for 15 min then blocked for 1 h at room temperature with 2 mL 2% Donkey serum, 1 mL Triton X-100, and 0.5 g BSA diluted in 42 mL of 1x PBS. After an hour, primary antibody was added and incubated overnight at 4 °C. Slides were then washed twice in PBS-T and once in PBS before adding secondary antibody and incubating for 1 h in the dark. Slides were again washed in PBS-T and PBS before adding Vectashield mounting media with DAPI.

## Behavioral characterization

WT and ΔLYST-B6 mice (male and female) were studied in eleven age groups (6 to 18 months of age; 5 to 10 animals per sex/strain/group).

Mice were assessed for ataxia and neurodegeneration phenotypes by scoring performance on seven screening tests: the ledge test, hindlimb clasping, gait, kyphosis, hindlimb locomotion, hindlimb stance, and ability to traverse a horizontal ladder and beam. Each test was assigned a score per defined performance criteria on a scale of 0–3, and the combined score was calculated across the seven tests. This composite score assesses coordination (ledge test, horizontal ladder, and beam), motor impairment (hindlimb clasping, hindlimb locomotion, hindlimb stance), kyphosis, and gait. Sample size information as follows, 6 mo: $n = 18$ ΔLYST-B6 (7 male, 11 female), $n = 17$ WT (9 male, 8 female); 7 mo: $n = 18$ ΔLYST-B6 (7 male, 11 female), $n = 17$ WT (9 male, 8 female); 8 mo: $n = 17$ ΔLYST-B6 (6 male, 11 female), $n = 16$ WT (9 male, 7 female); 9 mo: $n = 26$ ΔLYST-B6 (9 male, 17 female), $n = 18$ WT (5 male, 13 female); 10 mo: $n = 11$ ΔLYST-B6 (female only), $n = 7$ WT (female only); 11 mo: $n = 17$ ΔLYST-B6 (6 male, 11 female), $n = 16$ WT (9 male, 7 female); 12 mo: $n = 31$ ΔLYST-B6 (11 male, 20 female), $n = 31$ WT (17 male, 14 female), Heterozygous $n = 3$ (1 male, 2 female); 13-15 mo: $n = 6$ ΔLYST-B6 (male only), $n = 9$ WT (male only); 18 mo: $n = 7$ ΔLYST-B6 (3 male, 4 female), $n = 7$ WT (female only).

To further characterize motor coordination and balance, we performed three additional tests, including assessment of the gait by the inkblot footprint test, a horizontal balance beam test, and a vertical pole test.

For the inkblot footprint test, the front paws of 12-month-old mice were coated with non-toxic green ink and the hind paws coated in non-toxic black ink. Animals were then placed in a 60-cm long and 5-cm wide runway (with 30-cm clear, Perspex walls) with a small hut at the far end. A sheet of paper was placed on the bottom of the runway for each trial. Paw prints were analyzed as the average of four steps for (1) stance width, the average distance apart of the two front feet and the two hind feet, respectively; (2) stride length, the average distance of forward strides measured from the center of the middle paw pad on the same foot; (3) paw angle, the rotation of the paw away or towards the midline; and (4) stride angle, the angle between the midline of the mouse to the line between the hind paw and opposite front paw. Sample size information: $n = 10$ ΔLYST-B6 (3 male, 7 female), $n = 4$ WT (2 male, 2, female).

To test motor coordination and balance, a horizontal balance beam test was performed. Three days prior to testing, training sessions were performed on a 28-32-mm square beam to teach the mice how to compliantly run the length of the beam. At training day three, each mouse was able to traverse the beam for 4 consecutive trials. Mice were considered trained at the end of day 3 if they showed stable beam traversing behavior. Testing was conducted on two 80-cm long square beams, 24-mm and 12-mm wide, that were approximately 50-cm high with padding below the beam. The total time to traverse the beam and the total number of hind foot slips were averaged over three trials. Trial time was capped at 60 sec to traverse the beam. The total number of falls off the beam was also recorded. Sample size information as follows, 6 mo: $n = 18$ ΔLYST-B6 (7 male, 11 female), $n = 18$ WT (9 male, 9 female); 7 mo: $n = 18$ ΔLYST-B6 (7 male, 11 female), $n = 18$ WT (9 male, 9 female); 8 mo: $n = 18$ ΔLYST-B6 (7 male, 11 female), $n = 18$ WT (9 male, 9 female); 9 mo: $n = 25$ ΔLYST-B6 (9 male, 16 female), $n = 29$ WT (14 male, 15 female); 10 mo: $n = 17$ ΔLYST-B6 (6 male, 11 female), $n = 17$ WT (9 male, 8 female); 11 mo: $n = 17$ ΔLYST-B6 (6 male, 11 female), $n = 17$ WT (9 male, 8 female); 12 mo: $n = 31$ ΔLYST-B6 (11 male, 20 female), $n = 31$ WT (17 male, 14 female), $n = 3$ Heterozygous (1 male, 2 female); 13-15 mo: n = 6 ΔLYST-B6 (male only), $n = 9$ WT (male only); 18 mo: $n = 7$ ΔLYST-B6 (3 male, 4 female), $n = 7$ WT (female only).

Balance and coordination were also investigated using the vertical pole test. Briefly, mice were placed head up towards the top of a 50-cm high vertical pole with a diameter of 10 cm secured in the center of the home cage. The time taken for the animals to turn to a head downward position and the time to descend the pole after turning around were recorded and averaged over three trials. Sample size information as follows, 6 mo: $n = 18$ ΔLYST-B6 (7 male, 11 female), $n = 19$ WT (9 male, 10 female); 7 mo: $n = 18$ ΔLYST-B6 (7 male, 11 female), $n = 19$ WT (9 male, 810 female); 8 mo: $n = 17$ ΔLYST-B6 (6 male, 11 female), $n = 19$ WT (9 male, 10 female); 9 mo: $n = 14$ ΔLYST-B6 (9 male, 5 female), $n = 11$ WT (5 male, 6 female); 10 mo: $n = 17$ ΔLYST-B6 (6 male, 11 female), $n = 19$ WT (9 male, 10 female); 11 mo: $n = 17$

ΔLYST-B6 (6 male, 11 female), $n = 18$ WT (9 male, 9 female); 12 mo: $n = 31$ ΔLYST-B6 (11 male, 20 female), n = 32 WT (17 male, 15 female), $n = 3$ Heterozygous (1 male, 2 female); 13 mo: $n = 6$ ΔLYST-B6 (male only), n = 9 WT (male only); 14-15 mo: $n = 4$ ΔLYST-B6 (male only), $n = 9$ WT (male only); 18 mo: $n = 7$ ΔLYST-B6 (3 male, 4 female), $n = 7$ WT (female only).

WT and ΔLYST-DBA mice (male and female) were studied in five age groups, including 3, 4, 5, 6, and 7 months of age. Each experiment included 3-5 animals per sex/strain group. The ΔLYST-DBA model were subjected to the same behavioral characterization similar to the ΔLYST-B6 model.

### Cerebellar Purkinje cell quantification

The quantification of cerebellar Purkinje cells was adapted from a previous publication[10].

To determine the length of the Purkinje cell layer, a computer-assisted pipeline with researcher input was developed; all code is publicly available at https://github.com/mkhapp/Purkinje_Cell_Linear_Density/. Nuclear and calbindin stains were separated via color deconvolution utilizing QuPath 0.5.1. Next, using ImageJ-1.54i, the cerebellum granular layer was isolated from the nuclear-stained image using a top hat filter (radius 5), followed by a Gaussian blur filter (sigma 90), further isolated via Otsu thresholding, and eroded slightly to account for size increase due to blurring. This isolated layer was then presented to the supervising researcher for quality control. The outer perimeter of the granular layer was then measured (subtracting the distance between the start and end of the Purkinje cell layer). The linear density of the Purkinje cell layer was calculated by dividing the total number of manually counted Purkinje cells by the total length (mm) for each sample. The linear density is recorded as Purkinje cells per mm. Sample size information as follows, 3 mo: $n = 10$ ΔLYST-B6 (5 male, 5 female), $n = 10$ WT (5 male, 5 female); 6 mo: $n = 9$ ΔLYST-B6 (4 male, 5 female), $n = 10$ WT (5 male, 5 female); 9 mo: $n = 9$ ΔLYST-B6 (4 male, 5 female), $n = 9$ WT (4 male, 5 female), $n = 1$ Heterozygous (male); 12 mo: $n = 8$ ΔLYST-B6 (4 male, 4 female), $n = 7$ WT (5 male, 2 female), $n = 1$ Heterozygous (male); 15 mo: $n = 6$ ΔLYST-B6 (3 male, 3 female), $n = 4$ WT (female only); 18 mo: $n = 10$ ΔLYST-B6 (5 male, 5 female), $n = 10$ WT (5 male, 5 female). Slides from the ΔLYST-DBA mice were imaged and counted using the same methods as the ΔLYST-B6 slides described above.

### Transmission and whole-mount electron microscopy

Fixed sciatic nerve tissue samples were washed with 0.1 M cacodylate buffer (pH 7.4) buffer, post-fixed with 1% $OsO_4$ for 2 h, rinsed with 0.1 M cacodylate buffer then water, and placed in 1% uranyl acetate for 30 min. The nerve tissues were subsequently dehydrated in ethanol and propylene oxide and embedded in EMBed 812 resin (Electron Microscopy Sciences, Hatfield, PA). Approximately 80 nm thin sections were obtained with the Leica Ultracut-UCT Ultramicrotome (Leica Microsystems, Buffalo Grove, IL) and placed onto 300 mesh copper grids and stained with UranyLess (22409-20, Electron Microscopy Sciences, Hatfield, PA) and then with lead citrate.

Platelets were isolated by low-speed centrifugation of citrated plasma, washed briefly in PBS, and fixed in 2.5% glutaraldehyde for 30 min. Platelets in solution were then dropped onto grids and briefly dried. The grids were viewed with a JEM_1200_EXII Electron Microscope (JEOL Ltd, Tokyo, Japan) at 80 kV, and images were recorded on the XR611M 10. 5-megapixel, mid-mounted CCD camera (Advanced Microscopy Techniques Corp, Danvers, MA).

### Nerve fibril quantification

Analysis of peripheral sciatic nerves was performed by evaluating toluidine blue-stained slides. Each slide contained 4-7 cross sections of nerve and was imaged at 20X magnification and 1080 × 1080-pixel resolution. A blinded investigator manually counted the total number of nerve fibers in each cross-section of nerve. The fiber counts of each section were averaged and combined with other samples of the same strain to calculate the mean nerve fibers (expressed as mean ± SD). The total number of nerve fibers was plotted to evaluate the standard normal distribution for WT and ΔLYST-B6

mice. Analysis included 2 biological replicates of male ΔLYST-B6 and WT at 3- and 24- months of age with technical replicates of each sample (3 mo ΔLYST-B6 sample A: 4 replicates, 3 mo ΔLYST-B6 sample B: 9 replicates, 3 mo WT sample A: 22 replicates, 3 mo WT sample A: 12 replicates, 24 mo ΔLYST-B6 sample A: 6 replicates, 24 mo ΔLYST-B6 sample B: 5 replicates, 24 mo WT sample A: 12 replicates, 24 mo WT sample B: 11 replicates). Technical replicates were used for statistical analysis due to the limited resource availability of sectioned nerve tissue.

### Lipidomic analysis and RNA sequencing

Untargeted lipidomics of the left forebrain and cerebellar hemisphere of 18-month-old mice ($n = 3$ per strain) was done by Creative Proteomics (Shirley, NY). Normalized quantifications of individual lipids and lipid classes detected by mass spectrometry were grouped by brain area and genotype. Significant differences in abundances of both individual lipids and lipid classes between WT and ΔLYST-B6 mice were evaluated using a Two-Sample T-Test, with Bonferroni corrections to account for multiple testing in individual lipids. Results were summarized by sparse Partial Least Squares Discrimination Analysis (sPLS-DA) with the *mixOmics* R package[36]. Results from sPLS-DA and levels of individuals lipids and classes were plotted using the *ggplot2*, while heatmaps were plotted using the *gplots* R package.

RNA extraction from frozen right forebrain and cerebellar hemisphere, library preparation, and standard bulk RNA-sequencing (Illumina NovaSeq platform) were performed by Azenta Life Sciences. Reads were aligned to mm10 using HISAT2[37], and read counts were generated using DESeq2[38]. Differential expression was determined using the *results ()* function in the DESeq2[38] R package. All differentially-expressed genes (adjusted $P \le 0.05$) were then tested for functional enrichment using the WebGestaltR[39] R package. Metascape, an online gene and pathway annotation tool, was also used for comprehensive visualization of altered expression[40].

### Statistics and Reproducibility

All data visualization and statistical analyses were conducted in R. For all experiments, WT and ΔLYST-B6 mice were compared by either two-tailed Student's *t-test* assuming equal variances or two-tailed Student's *t-test* assuming unequal variances. Data are presented as the mean ± 1 SE with a $P$ of $\le 0.05$ considered statistically significant for individual comparisons. For comparisons with multiple testing considerations, a Bonferroni correction was used to set the significance threshold.

The sample size was determined by resource availability and feasibility of experiments. The sample size of animals for various experiments was deemed sufficient to demonstrate reproducibility of acquired results and to provide sufficient data for calculated power while also considering ethical use and treatment of experimental animals. Experimental groups were determined by animal genotype; however, covariates were controlled by using inbred transgenic animals. The qPCR and Western Blot experiments were repeated with consistent results across the tissue types of 3–6 animals per sex/strain. Experiments, including behavioral assays and histological analyses, were repeated in separate trials with consistent results across cohorts of 5–10 animals per sex/strain/age group.

Histological images of tissue that were severely damaged in the preparation of tissues and slides were excluded from quantification (nerve fibril/Purkinje cells) data analysis. This exclusion was to reduce the influence of artifacts and tissue preparation in statistical analysis. During behavioral testing using the balance beam, animals who scooted across the beam were excluded from analysis due to the inappropriate crossing of the beam not accepted by the methodology. Nerve fibril quantification and Purkinje cell quantification was performed by a blinded investigator. Histological imaging of the brain, retina, and blood smear was performed by a blinded investigator. Investigator blinding was not possible for behavioral testing due to the visually distinct phenotypic coat color of the mutant mice.

### Reporting summary

Further information on research design is available in the Nature Portfolio Reporting Summary linked to this article.

**Article**

## Data availability
All data supporting the findings of this study are available within the paper and Supplementary Information. All data and accompanying information for the transcriptomic and lipidomic datasets used in this study is publicly available and deposited in Gene Expression Omnibus (GEO accession number GSE266142). Numerical data for enrichment, P value, and FDR from RNA-seq analysis of the forebrain and cerebellum in 3- and 18-month-old WT and ΔLYST-B6 mice can be found in Supplementary Data 3. Source data for figures can be found in Supplementary Data 5.

## Code availability
All core analyses were performed using publicly available software as mentioned in the Methods. All data visualization and statistical analyses were performed using R (version 4.2.3). Complete R scripts for processing and plotting (e.g., principal component analysis, volcano plots, heatmaps) are available from the corresponding author upon request.

## Abbreviations
| | |
|---|---|
| CHS | Chediak-Higashi syndrome; |
| LYST | Lysosomal Trafficking Regulator |
| IF | immunofluorescence |
| IHC | immunohistochemistry |
| WT | Wild Type |
| PMN | polymorphonuclear leukocytes |
| MEF | Mouse Embryonic Fibroblasts |
| TEM | transmission electron microscopy |
| qPCR | quantitative polymerase chain reaction |
| PD | Parkinson's Disease |
| AD | Alzheimer's Disease |

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

## Acknowledgements
This study is partly supported by the Intramural Research Programs of the National Human Genome Research Institute (NHGRI) from the National Institutes of Health. The authors would like to thank Heidi Dorward and Stephen Wincovitch (NHGRI) for their technical assistance in acquiring microscopy images, and Dr. Joseph Brzostowski (NIAID) for facilitating image processing and analysis.

## Author contributions
S.G. and M.L.T. acquired, analyzed, and interpreted data, visualized data, and drafted the manuscript. F.G.F. performed computational analyses for lipidomics and transcriptomics and assisted in data interpretation and visualization. J.D.B., D.Y. and E-R.N. generated and established the animal model and performed genetic sequencing and I.F. of MEFs. G.E. and L.G. generated and maintained the model. D.M. and M.M. performed protein expression analysis. P.L. performed IF of the brain and confocal imaging. P.M.Z. performed transmission and whole-mount EM and tissue IHC. D.S. and A.N. performed and analyzed behavioral phenotyping. M.T. performed image processing and quantified Purkinje cell layer length. W.A.G., W.J.I. and M.C.V.M. conceptualized and supervised the study. All authors reviewed the manuscript.

## Funding

## Competing interests
The authors declare no competing interests.
