## [Transparent Peer Review file · Communications Biology]

A murine model lacking *Lyst* recapitulates Chediak-Higashi syndrome with an early-onset neurodegenerative phenotype

Corresponding Author: Dr May Christine Malicdan

Version 0:

Reviewer comments:

Reviewer #1

(Remarks to the Author)

The manuscript by Greene et al. reports a new *Lyst* mutant mouse strain which develops an important feature (early onset neurological defects) which was absent from previous *Lyst* models. In humans, *LYST* mutations cause Chediak-Higashi syndrome, a disease characterized by immunodeficiency (that is treatable by bone-marrow transplantation) and progressive neurological impairment (that is currently untreatable). The new model was used to study the neurological features of the disease and will likely be a valuable tool for the community in testing new therapies.

This is a very well-prepared manuscript (solid data, careful methods, flawless grammar, rigorous statistics, consideration of limitations, etc.) that advances study of Chediak-Higashi syndrome in several ways (new information of lipidomics, new hypotheses on microglia, new model characterization). There are no major flaws, but a few requests to address and comments to consider.

The following are minor points that need to be addressed:

1. Lines 405-407 should be moved from the Results to the Discussion. It might also be considered to drop this text, it's not rigorous to assume that a greater number of changes means "more important". Alternatively, given the ubiquitous expression of *Lyst*, it might be explored further in the Discussion to fortify the existing text in L488-493.
2. In Table 1, ophthalmic features of D2.*Lyst*^{bg-J} mice are described in PMID: 20617205.
3. In Figure 2b, recommend using the label "Retina" instead of "Eye".
4. Please double-check the labeling of the RPE-Choroid-Sclera in Figure 2b WT – the RPE is the thin pigmented band, the choroid the next pigmented layer (thicker and less dense), choroid the next layer (fibrous, non-pigmented). The additional layer in WT, not present in *LYST*, is likely muscle based on the nuclei. It appears that "8", "9", and "10" need to be nudged up in both panels (with 10 added to *LYST*, and perhaps the extra layer only in WT explained or labeled as muscle).
5. In Line 320, the altered pigmentation is likely in the choroid. Also, note that it's not obvious if there is less pigmentation, or a change in the distribution of how the pigment is distributed (both at the tissue level, which seems to be thickened, or at the organelle level, with the *Lyst* mutant melanosomes presumably being enlarged. "altered pigmentation" (or a phrase like it) might be preferable to "reduced pigmentation".
6. The omics data must be deposited into a public repository, preferably with an associated spreadsheet making the full analyzed data set readily accessible to non-bioinformaticists.
7. Please make sure that mutation name is registered with the MGI data repository, and if changes are needed to keep with nomenclature conventions, make sure it matches the use in the paper and ancillary data documents deposited elsewhere.
8. In the Methods, please describe whether the controls were inbred mice (likely housed separately etc) or littermate controls.

The following are comments which the authors are encouraged to consider, and may want to incorporate into the manuscript:

1. It seems interesting and important that the same transcriptional changes you emphasize here, especially *C3ar1* and *Tyrobp*, were previously identified in a microarray study of the *Lyst* mutant iris (PMID: 20739468). From GSE16994, using the simple Geo2R analysis provided, these are both in the top 20 DEGs, changing in the same direction as observed here – several complement genes are as well. The implication might be that pigmented cells are being insulted in the same manner

as neurons. A PDF of this critique, which also includes image exports from GEO, is attached

2. The "Ophthalmic phenotypes" (row title in Table 1) that have been studied vary a bit from lab to lab, with the Anderson lab largely focused on the iris and other labs mostly on the retina. Not sure there is space to delineate here but pointing it out for your consideration.

Dr. Michael Anderson
University of Iowa

Reviewer #2

(Remarks to the Author)

In this manuscript Greene et al report a novel model of Chediak-Higashi syndrome. Chediak-Higashi syndrome is caused by mutation in the lysosomal trafficking regulator gene LYST. CRISPR-Cas9 has been used to create a model lacking the murine Lyst gene. Validation of the model shows reduced Lyst mRNA and absent Lyst protein across numerous tissue tested. The authors have characterised this mouse and found several interesting phenomenon.

Δ LYST mice recapitulate the bleeding deficiencies seen in CHS, and also present with the CHS characteristic enlarged granules in leukocytes and monocytes.

Motor behavioural deficits were evident in the mice from 6 months of age.

IHC revealed that the Purkinje cell layer in the Δ LYST mice shows areas of cell loss by 12 months, however this decrease is only statistically significant at 18 months.

Interestingly, already at 3 months of age the Δ LYST mice have myelin decompaction of sciatic nerves, which worsens with age. Significant loss of nerve fibers is already evident at 3 months of age in Δ LYST mice.

The authors also present data on lipidomic analyses of 18 month old mice, and discuss broad lipid dysregulation.

Bulk-RNA seq transcriptomic analyses are conducted in 3 and 18 month old cerebellum and forebrain, and DEGs discussed.

The lipidomic and transcriptomic data is descriptive, however these datasets are likely to be of use for downstream mechanistic insight and important for the field.

Overall, this work is important as current mouse models recapitulate several disease features, but none present with an overt or severe neurological deficit. Thus this mouse model presents an advancement in modelling of disease, as it presents with early and significant neurological deficits. The characterisation of this model is robust and thorough, and an interesting read.

I recommend this manuscript for publication, following minor revision (see comments below),

Emma Clayton

Minor comments-

The methods mention a 12 hr light/dark cycle (from 06.00 to 14.00). Is this a typo?

Unclear to me how the colony were maintained- in a homozygous state? If so, in the behavioural assays are the WT mice littermates or not?

Statistical significance is only seen at 18 months in the Purkinje cell layer in the CHS mouse model- the results in the text should be rephrased to reflect this. The median decrease in the earlier ages is not statistically significant, and thus should be reported as a trend.

It is unclear in the text what age mice were used for the lipidomic study (although it is clear in the figure legend), this information should be added to the main text.

Version 1:

Reviewer comments:

Reviewer #1

(Remarks to the Author)

Thank you for your consideration of my prior comments, which have been appropriately addressed.

There is one grammatical issue you might correct with the editor's permission. In the new sentence in Methods describing the controls ("Controls were inbred WT C57BL/6J, originally derived from the heterozygous in-crosses and maintained in parallel with the homozygous line", it seems a cut/paste issue led to "inbred" being incorrectly inserted prior to "C57BL/6J". Obviously, the mice derived from a cross with your CRISPR treated mice aren't inbred, just derived from a cross with inbred mice, so it has little likelihood of being confusing to readers - only something worth cleaning up if possible.

As previously shared, this is a very well-prepared manuscript (now even more so) that advances study of Chediak-Higashi syndrome in several ways.

Reviewer #2

(Remarks to the Author)

I am happy that the authors have addressed my minor comments, and recommend that this manuscript is published.

May Christine V. Malicdan, M.D., Ph.D.
Director, UDP Translational Research Laboratory
NIH Undiagnosed Diseases Program
Associate Investigator, Medical Genetics Branch
National Human Genome Research Institute, National Institutes of Health

10 Center Drive - MSC 1851
Building 10, Room 10C103
Bethesda, Maryland 20892-1851
Office: (301) 451 4665 | Cell: (301) 272 0599
Email: maychristine.malicdan@nih.gov

December 30, 2024

Dear Reviewers,

Many thanks for taking the time to review our manuscript. Your feedback was helpful in improving this work.

Reviewer #1:

Minor Comments

Lines 405-407 should be moved from the Results to the Discussion. It might also be considered to drop this text, it's not rigorous to assume that a greater number of changes means "more important". Alternatively, given the ubiquitous expression of Lyst, it might be explored further in the Discussion to fortify the existing text in L488-493.

Response: We agree these lines are better served in the Discussion. We have moved to line 475 to support further discussion on the role of neuroinflammation in neurodegeneration. We agree with Reviewer #1 that a greater number of differentially expressed genes (DEGs) is not necessarily more important. We would like to convey that the finding of a lack of immune pathway DEGs in younger mice provides interesting considerations regarding the activation of neuroimmune pathways over the course of disease. Thus, these lines have been moved to fortify existing text in the discussion.

In Table 1, ophthalmic features of D2.Lystbg-J mice are described in PMID: 20617205.

Response: Thank you for bringing this additional publication to our attention. This adds to our exploration of the lipidomic changes in the brain through the lens of understanding oxidative membrane damage due to poor repair of lipid membranes in pigmented cells of the iris. This article also corroborates lipidomic changes with the identification of elevated lipid hydroperoxides in D2.Lystbg-J mice. In addition, it highlights the sensitivity of genetic background to disease severity. We have added this reference to Table 1 and added to the bibliography.

In Figure 2b, recommend using the label "Retina" instead of "Eye".

Response: We agree with the reviewer that labeling as "Retina" is more appropriate than "Eye." We have altered the figure legend to reflect this change.

Please double-check the labeling of the RPE-Choroid-Sclera in Figure 2b WT – the RPE is the thin pigmented band, the choroid the next pigmented layer (thicker and less dense), choroid the next layer (fibrous, non-pigmented). The additional layer in WT, not present in LYST, is likely

muscle based on the nuclei. It appears that “8”, “9”, and “10” need to be nudged up in both panels (with 10 added to LYST, and perhaps the extra layer only in WT explained or labeled as muscle).

Response: Thank you for bringing this to our attention! We have clarified the labeling of the figure and the figure legend to reflect the correct anatomical layers.

In Line 320, the altered pigmentation is likely in the choroid. Also, note that it's not obvious if there is less pigmentation, or a change in the distribution of how the pigment is distributed (both at the tissue level, which seems to be thickened, or at the organelle level, with the Lyst mutant melanosomes presumably being enlarged. “altered pigmentation” (or a phrase like it) might be preferable to “reduced pigmentation”.

Response: We agree that a more appropriate assessment of the pigment changes is either “altered” or “atypical.” This has been changed to “altered pigmentation” to better reflect the findings on Line 322.

The omics data must be deposited into a public repository, preferably with an associated spreadsheet making the full analyzed data set readily accessible to non-bioinformaticists.

Response: The -omics data has been deposited into the Gene Expression Omnibus (GEO). This has been added to Data Availability statement on Line 540- 543.

Please make sure that mutation name is registered with the MGI data repository, and if changes are needed to keep with nomenclature conventions, make sure it matches the use in the paper and ancillary data documents deposited elsewhere.

Response: The mutation name and information has been registered with the MGI data repository with appropriate nomenclature conventions.

In the Methods, please describe whether the controls were inbred mice (likely housed separately etc) or littermate controls.

Response: Thank you for improving manuscript by clarifying this methodology. The knockout colony was derived from het-het crosses and maintained in a homozygous state via viable homozygous crosses. Wildtype controls were inbred mice initially derived from the het-het crosses, and expanded in parallel. As such, these WT mice were housed separately and are not littermates. This additional clarification has been added in the methods section on Line 123-124.

Considerations

It seems interesting and important that the same transcriptional changes you emphasize here, especially C3ar1 and Tyrobp, were previously identified in a microarray study of the Lyst mutant iris (PMID: 20739468). From GSE16994, using the simple Geo2R analysis provided, these are both in the top 20 DEGs, changing in the same direction as observed here – several complement

genes are as well. The implication might be that pigmented cells are being insulted in the same manner as neurons. A PDF of this critique, which also includes image exports from GEO, is attached .

Response: This brings further insights to potential disease mechanisms occurring in both neurons and pigmented cells caused by variations in *Lyst*. Interestingly, melanocytes, glial cells, neurons, and peripheral nervous system neurons are all derived from the neural crest¹. The same pattern of transcriptomic changes, such as in of *C3ar1* and *Tyrobp*, in B6.*Lyst*^{bg-J} irises² and in the brain identified here opens new questions regarding the role of *Lyst* in neural crest derived cells. However, exploring these interesting patterns and their implications is outside the scope of this manuscript. Future studies may evaluate the shared cellular processes of neural crest derived cells that require *Lyst* and potentially identify ways to circumvent these cellular insults to treat disease.

1. Sandell LL, Trainor PA. Neural Crest Cell Plasticity: Size Matters. In: Madame Curie Bioscience Database [Internet]. Austin (TX): Landes Bioscience; 2000-2013. Available from: <https://www.ncbi.nlm.nih.gov/books/NBK6337/>
2. Trantow CM, Cuffy TL, Fingert JH, Kuehn MH, Anderson MG. Microarray analysis of iris gene expression in mice with mutations influencing pigmentation. Invest Ophthalmol Vis Sci. 2011 Jan 5;52(1):237-48. doi: 10.1167/iovs.10-5479. PMID: 20739468; PMCID: PMC3053276.

The “Ophthalmic phenotypes” (row title in Table 1) that have been studied vary a bit from lab to lab, with the Anderson lab largely focused on the iris and other labs mostly on the retina. Not sure there is space to delineate here but pointing it out for your consideration.

Response: We recognize the ophthalmic phenotypes is a larger generalization of the changes identified in *Lyst* models across various laboratories. Ophthalmic findings in Chediak-Higashi Syndrome are complex with changes in the iris¹, retina², and optic nerves³. In this manuscript, we aim to take a big picture snapshot of limitations and benefits of each known mouse model of Chediak-Higashi Syndrome. We added some clarification in the table and also included a footnote to delineate that most studies were focused on the retina, except for Trantow *et al* that focused on the iris.

1. Trantow CM, Mao M, Petersen GE, Alward EM, Alward WL, Fingert JH, Anderson MG. *Lyst* mutation in mice recapitulates iris defects of human exfoliation syndrome. Invest Ophthalmol Vis Sci. 2009 Mar;50(3):1205-14. doi: 10.1167/iovs.08-2791. Epub 2008 Nov 21. PMID: 19029039; PMCID: PMC2693230.
2. Ji X, Zhao L, Umaphathy A, Fitzmaurice B, Wang J, Williams DS, Chang B, Naggert JK, Nishina PM. Deficiency in *Lyst* function leads to accumulation of secreted proteases and reduced retinal adhesion. PLoS One. 2022 Mar 3;17(3):e0254469. doi: 10.1371/journal.pone.0254469. PMID: 35239671; PMCID: PMC8893605.
3. Desai N, Weisfeld-Adams JD, Brodie SE, Cho C, Curcio CA, Lublin F, Rucker JC. Optic neuropathy in late-onset neurodegenerative Chediak-Higashi syndrome. Br J Ophthalmol. 2016 May;100(5):704-7. doi: 10.1136/bjophthalmol-2015-307012. Epub 2015 Aug 25. PMID: 26307451.

Reviewer #2

Minor Comments

The methods mention a 12 hr light/dark cycle (from 06.00 to 14.00). Is this a typo?

Response: Thank you for identifying this error! This was a typo and the light/dark cycle time has been corrected to 06:00 to 18:00.

Unclear to me how the colony were maintained- in a homozygous state? If so, in the behavioural assays are the WT mice littermates or not?

Response: Thank you for improving manuscript by clarifying this methodology. The knockout colony was derived from het-het crosses, then maintained in a homozygous state via viable homozygous crosses. Wildtype controls were inbred mice, initially derived from the het-het crosses and maintained in parallel. As such, these WT mice were housed separately and are not littermates. This additional clarification has been added in the methods section on Line 123-124.

Statistical significance is only seen at 18 months in the Purkinje cell layer in the CHS mouse model- the results in the text should be rephrased to reflect this. The median decrease in the earlier ages is not statistically significant and thus should be reported as a trend.

Response: We agree with the Reviewer's comment to rephrase and have clarified statistical significance only at 18 months-of-age in results section at line 360.

It is unclear in the text what age mice were used for the lipidomic study (although it is clear in the figure legend), this information should be added to the main text.

Response: We are glad to provide more clarification here and in the manuscript. Lipidomic analysis was performed in 18-month-old WT and mutant mice ($n=3$). This information has been clarified in the Methods on Line 278 and Results sections on Line 376.

May Christine V. Malicdan, M.D., Ph.D.
Director, UDP Translational Research Laboratory
NIH Undiagnosed Diseases Program
Associate Investigator, Medical Genetics Branch
National Human Genome Research Institute
National Institutes of Health

10 Center Drive - MSC 1851
Building 10, Room 10C103
Bethesda, Maryland 20892-1851
Office: (301) 451 4665
Cell: (301) 272 0599
Email: maychristine.malicdan@nih.gov

April 7, 2025

Thank you for considering our manuscript “A Murine Model Lacking Lyst Recapitulates Chediak-Higashi Syndrome with an Early-Onset Neurodegenerative Phenotype” for publication in *Communications Biology*.

We have read the final minor Reviewer’s comment and we appreciate the opportunity to revise as required. improve our manuscript. Please find below a point-by-point response. We are submitting the revised manuscript as a tracked version and one with an article file.

Reviewer #1:

Thank you for your consideration of my prior comments, which have been appropriately addressed.

There is one grammatical issue you might correct with the editor's permission. In the new sentence in Methods describing the controls ("Controls were inbred WT C57BL/6J, originally derived from the heterozygous in-crosses and maintained in parallel with the homozygous line", it seems a cut/paste issue led to "inbred" being incorrectly inserted prior to "C57BL/6J". Obviously, the mice derived from a cross with your CRISPR treated mice aren't inbred, just derived from a cross with inbred mice, so it has little likelihood of being confusing to readers - only something worth cleaning up if possible.

As previously shared, this is a very well-prepared manuscript (now even more so) that advances study of Chediak-Higashi syndrome in several ways.

Response: Many thanks for the kind words and we have revised that part of the Methods accordingly, and wrote “Controls were WT C57BL/6J, originally derived from the heterozygous in-crosses and maintained in parallel with the homozygous line.”

Thank you for considering our manuscript for publication.

Sincerely,

May Christine V. Malicdan, M.D., Ph.D.